# Complexin-1 regulated assembly of single neuronal SNARE complex revealed by single-molecule optical tweezers

Tongrui Hao [1,2✉], Nan Feng[1,2], Fan Gong[3], Yang Yu[3], Jiaquan Liu [1✉] & Yu-Xuan Ren [4✉]

The dynamic assembly of the Synaptic-soluble N-ethylmaleimide-sensitive factor Attachment REceptor (SNARE) complex is crucial to understand membrane fusion. Traditional ensemble study meets the challenge to dissect the dynamic assembly of the protein complex. Here, we apply minute force on a tethered protein complex through dual-trap optical tweezers and study the folding dynamics of SNARE complex under mechanical force regulated by complexin-1 (CpxI). We reconstruct the clamp and facilitate functions of CpxI in vitro and identify different interplay mechanism of CpxI fragment binding on the SNARE complex. Specially, while the N-terminal domain (NTD) plays a dominant role of the facilitate function, CTD is mainly related to clamping. And the mixture of 1-83aa and CTD of CpxI can efficiently reconstitute the inhibitory signal identical to that the full-length CpxI functions. Our observation identifies the important chaperone role of the CpxI molecule in the dynamic assembly of SNARE complex under mechanical tension, and elucidates the specific function of each fragment of CpxI molecules in the chaperone process.

[1] State Key Laboratory of Molecular Biology, Shanghai Institute of Biochemistry and Cell Biology, Center for Excellence in Molecular Cell Science, Chinese Academy of Science, Shanghai 200031, China. [2] University of Chinese Academy of Sciences, Beijing 200049, China. [3] National Facility for Protein Science in Shanghai, Zhangjiang Lab, Shanghai Advanced Research Institute, Chinese Academy of Sciences, Shanghai 201210, China. [4] Institute for Translational Brain Research, Shanghai Medical College, Fudan University, Shanghai 200032, China. ✉email: haotongrui2018@sibcb.ac.cn; liujiaquan@sibcb.ac.cn; yxren@fudan.edu.cn

Vesicle transport is involved in a plethora of important cell activities, such as neurotransmitter release, hormone secretion, and intracellular transport. Misfunction of the vesicle transport leads to organelle defect and cell dysfunction. It correlates with the occurrence and development of many diseases such as neurodegenerative diseases, diabetes, infection and immune deficiency. The researches on vesicle transport were repeatedly recognized by Nobel prizes. Despite the importance, the understanding of the complex and delicate intracellular transport is still preliminary and tentative, and many of the finer regulatory mechanisms of vesicle transport remain to be further elucidated. Among them, nerve cells are the most representative in vesicle transport, owing to the existence of a special type of vesicle in nerve cells, namely synaptic vesicles, which participate in the release of neurotransmitters.

The neurotransmitter release and intercellular communication requires the membrane fusion, which takes place within a millisecond[1]. The membrane fusion is driven by crucial fusion proteins, such as the molecular machine Synaptic-soluble N-ethylmaleimide-sensitive factor Attachment REceptor (SNARE) proteins. Neuron SNARE includes three kinds of monomer proteins: Syntaxin, SNAP-25 (Synaptosome-associated protein 25 kDa) and VAMP (Vesicle-Associated Membrane Protein)[2]. SNARE has a typical heptanucleotide repeating domain (SNARE motif), in which different SNARE monomer proteins contribute a central arginine (R) or glutamyl (Q) residues at the ionic layer. So, the SNARE monomers are divided into R- or Q-SNARE, respectively[3]. VAMP is R-SNARE and has a SNARE motif. Snap-25 and Syntaxin-1 are both Q-SNARE, with two and one SNARE motif respectively. In the SNARE hypothesis[4], different vesicles have different "vesical-SNARE" (V-SNARE, namely VAMP), and the target members have the "target-SNARE" (T-SNARE, syntaxin and SNAP-25). Only when the correct V-SNARE and T-SNARE recognize each other can the SNARE complex be correctly assembled to drive the fusion of vesicles and plasma membranes.

Meanwhile, many regulatory proteins, e.g., N-ethylmaleimide sensitive factor (NSF), soluble NSF adaptor proteins (SNAPs), Complexin (Cpx) and synaptotagmin-1, are involved in the regulation of SNARE zippering and are crucial for the membrane fusion to efficiently occur at the precise time in vivo[4,5]. In the absence of SNAP and NSF, the three SNARE monomers combine to form a stable four-helix bundle, i.e., SNARE complex[5,6]. Traditional ensemble approaches are not able to dissect the dynamic assembly of SNARE, and are insufficient to record the less populated misassembled states[7,8]. In addition, functional SNARE assembly occurs in the presence of the opposing force imposed by negatively charged membranes, which has a great impact on the kinetics and regulation of SNARE assembly[9,10].

However, the functions of many important regulators have not been well understood in single-molecule level. Single-molecule optical tweezers can apply mechanical force and measure the distance in real-time with high precision and low photodamage on biological molecules. Although the single neuronal SNARE complex zipper in three distinct stages with N-terminal crosslinked[9], it is still unclear on how the dynamic SNARE assembly is regulated by many proteins, e.g., complexin. As a small cytosolic α-helical protein[11], Cpx can bind on the SNARE complex, and executes unique functions[12]. We use single-molecule optical tweezers to study the Cpx-regulated SNARE assembly, and dissect the dynamic conformational change and force-dependent intermediate states. We found that by crosslinking near the site at -6 layer (by the disulfide bonds between VAMP2 (Q36C) and syntaxin 1 (L209C)), the SNARE complex maintained a four-state transition at constant trap separation mode and the SNARE complex suggested distinguishable

dynamics in presence of various fragments of CpxI molecules owing to respective binding sites from single-molecule experiment. The interplay of the CpxI with individual functional SNARE complex would be identified through dual-trap optical tweezers during dynamic transition under mechanical tension[13,14]. Such single molecule technique is not only a tool complementary to the ensemble methods, but also can detect many important and less populated intermediates involved in the protein function. Specifically, in the interplay of CpxI and SNARE assembly, distinguishable motifs of the CpxI molecules suggests totally different binding sites and configurations revealed by our single-molecule approach.

## Results

**Single-molecule study with dual-trap optical tweezers.** We built a dual-trap optical tweezers on a concrete background in the Molecular Imaging System in National Facility for Protein Sciences Shanghai, and adopted differential detection to minimize the systematic noise coupled from the environment[14–16]. The instrument was also isolated from all the environmental noise including the fan from the laser controller, and all the operation were performed outside the isolated room after the samples were loaded to the microfluidics. The high-resolution dual-trap optical tweezers were built by splitting a 1064 nm infrared laser (Spectra Physics) with orthogonal polarization (Layout in Fig. 1a). Two microspheres were captured by the dual-trap optical tweezers. One microsphere coated with the protein-DNA complex is kept stationary, while the other with streptavidin coating is movable to approach the stationary microsphere to form a tether (See supplementary movie for demonstration of the fishing process, Supporting information 6). Despite movable, the beam spot on the back focal plane of the objectives is stationary with intensity distribution depending on the trapped microsphere position. Once the tether is detected, the movable trap separates from the stationary trap to a certain distance to apply force on the tether of the DNA handle connected with the protein.

To observe reversible and regulatory SNARE assembly, we designed SNARE complexes containing the full cytoplasmic domain and a crosslinking site between syntaxin and VAMP2 (synaptobrevin-2) near the -6 hydrophobic layer (Fig. 2b). The SNARE proteins were purified independently, then preassembled into SNARE complexes in vitro (Supplementary Figure 2). Our CpxI-SNARE binding experiment corroborated that CpxI does not bind to any SNARE monomers, but do bind to the SNARE complex (Supplementary Figure 3). A 2,260-bp DNA handle (Supporting information 5) containing an activated thiol group at its 5′end was added to the solution of SNARE complex with a molar ratio of 1:20. Intramolecular and intermolecular crosslinking occurred in open air between the cysteine residues on syntaxin and VAMP2 (intramolecular) and between VAMP2 and the DNA handle (intermolecular). The DNA handle contains two digoxigenin moieties at the 3′ end. Both the thiol group and digoxigenin moieties on the handle were introduced in the PCR through primers. During the single-molecule experiment, the SNARE complex was tethered through DNA handle between anti-dig and streptavidin beads through the dual-dig and biotin tag (Fig. 1b).

To manipulate a single SNARE complex, we either pulled or relaxed the complex by moving one optical trap relative to the other at a constant speed (typically of 20 nm/s) or held the complex under constant trap separation. Both force and extension of the protein-DNA complex were recorded at 10 kHz. We had tried to crosslink the SNARE at four sites (−1, −2, −3, −6 layer, Supplementary Figure 4, Supplementary Figure 5, Supporting information 8), however, only the

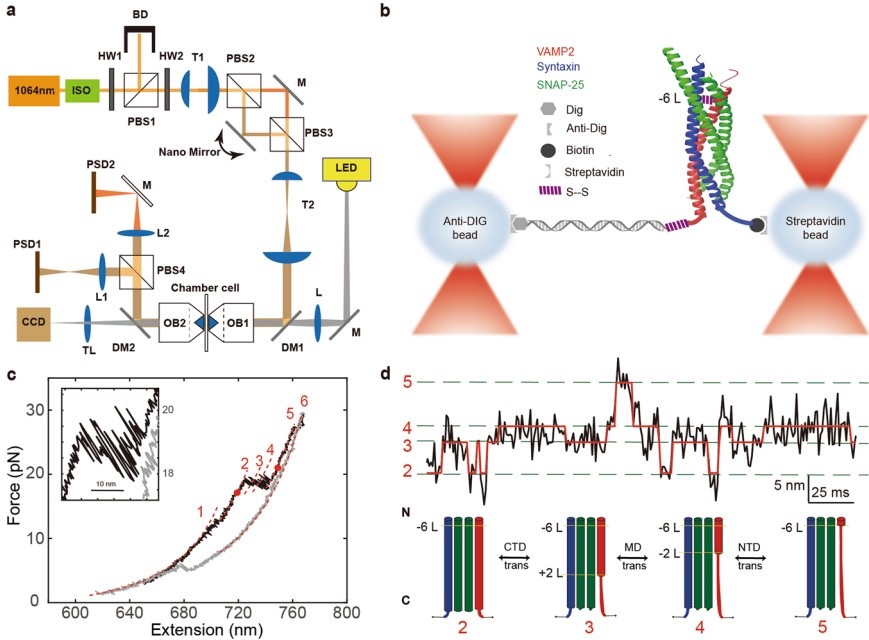

**Fig. 1 Schematics of dual-trap optical tweezers for single molecule study. a** Layout of dual-trap optical tweezers. The trapping beam comes from a solid-state laser with central wavelength of 1064 nm. ISO, isolator; HW, half-wave plate; BD, Beam Dump; T, telescope; PBS, polarizing beam splitter; OBJ, water immersion objective; L, lenses; TL, tube lens; PSD, position sensitive detector; LED, laser emitting diode; DM, dichroic mirror. **b** Schematics of the single molecule essay. The preassembled SNARE complex was crosslinked to a 2,260-bp DNA handle through disulfide bond and further attached with two microspheres through either digoxigenin/anti-digoxigenin or biotin/streptavidin. **c** Force-extension curves (FECs) of a single SNARE complex during pulling (black) and relaxing (gray). The continuous regions of the FECs corresponding to different assembly states (marked by red numbers) were fitted using the worm-like chain model (red lines). The inset shows a close-up view of the region between two red-dot markers. **d** Extension-time trajectories of single SNARE complexes under constant trap separation. The ideal state transitions derived from hidden-Markov model (HMM) are expressed in red lines. The positions of different states are marked by green dashed lines and labeled with the state numbers (Supplementary Figure 1, Supplementary Table 1). Data were filtered using a time window of 0.2 ms. SNARE configurations correspond to the states in extension-time trajectories, black numbers and L (−6 L, −2 L, +2 L) indicate different layers (−6 layer, −2 layer, +2 layer).

crosslinking at two of them (−2, and −6 layer) were successful in the subsequent single molecule experiment as the probability to form an effective tether was very low for the constructs crosslinked at either −1 or −3 layers. In order to let the CpxI interact sufficiently with longer NTD fragments in SNARE, most of the data assumes a crosslink at the −6 layer, such that the interaction region on the SNARE complex is maximum. When the force rises to 7~13 pN, we observe a ~3 nm slow hopping (between states 1 and 2 in Fig. 1c) corresponding to the opening of the linker domain of the SNARE complex[9,13]. The SNARE transits from fully zippered/folded state 1 to linker-open state 2 when the linker domain is disassembled. However, when the force increases to 17~19 pN[9], the SNARE complex hops among four states (state 2-5) with maximum extension difference of ~15 nm (Fig.1c) consistent with the cross-link at −6 layer[9,13,18]. Thus, the complete dissociation of the SNARE complex at the middle domain (MD, state 3-4) and N-terminal (state 4-5) does not occur immediately after the assembly of the SNARE complex at the C-terminal (state 2-3), because of the existence of a nucleation site for NTD zippering at N-terminal to the −6 layer[13]. So the transition among States 2-5 is different from the hopping+rip signal of −7 layer cross-link (or named as N-terminal cross-link) SNARE complex[9]. In this case, the slow dissociation of SNARE complex's four helix bundle is represented by the continuous hopping (between state 2 to 5 in Fig. 1c). Notice that there is a dissociation of the Middle domain (MD) of the SNARE complex (states 3-4) near the central ion layer (0 layer). When the force continues to increase, a rip appears for the drop of SNAP25 which is not linked by disulfide bond. When the force gradually decreases below 5 pN, the SNARE complex could not return to

full assembly due to the absence of SNAP25 (Fig.1c, gray curve). The Force-extension curve of the handle DNA would be fitted using Worm-like chain model[19] (dashed curve in Fig. 1c). Fig 1d illustrates a typical extension-time trace when the trap separation is kept fixed with an applied average force of ~19 pN. The SNARE complex would transit among 4 states at equilibrium (state 2-5, Fig. 1d), which correspond to the four different conformations of SNARE complex: linker domain opens (State 2), C-terminal unfolds to +2 layer (State 3), MD unfolds to −1 layer (State 4), and N-terminal unfolds before SNAP-25 drop (State 5).

**CpxI clamp partially folded SNARE complex.** To characterize CpxI-dependent SNARE assembly/disassembly, we measured the extension-time trajectories at a certain average force where SNARE transits among 4 states (Fig. 2a). Firstly, the captured SNARE complex was stretched for a full cycle to check correct assembly (See supplementary video for the fishing process) and to determine the equilibrium force of intermediate state. The average equilibrium force is generally between 18 and 20 pN. The overall distribution of different SNARE molecules' equilibrium force approximates a normal distribution for all the measured SNARE complex. An individual SNARE complex was held by dual-trap optical tweezers with a fixed trap separation, and 8 μM CpxI was supplied in a separate channel ('protein channel', Supplementary Figure 6) to directly deliver CpxI molecules to the region where the SNARE folding/unfolding took place. Cpx consists of four domains, N-terminal domain (NTD, 1-26 aa), accessory helix (AH, 26-48 aa), center helix (CH, 48-70 aa) and C-terminal domain (CTD, 70-134 aa) (inset in Fig. 2a). The

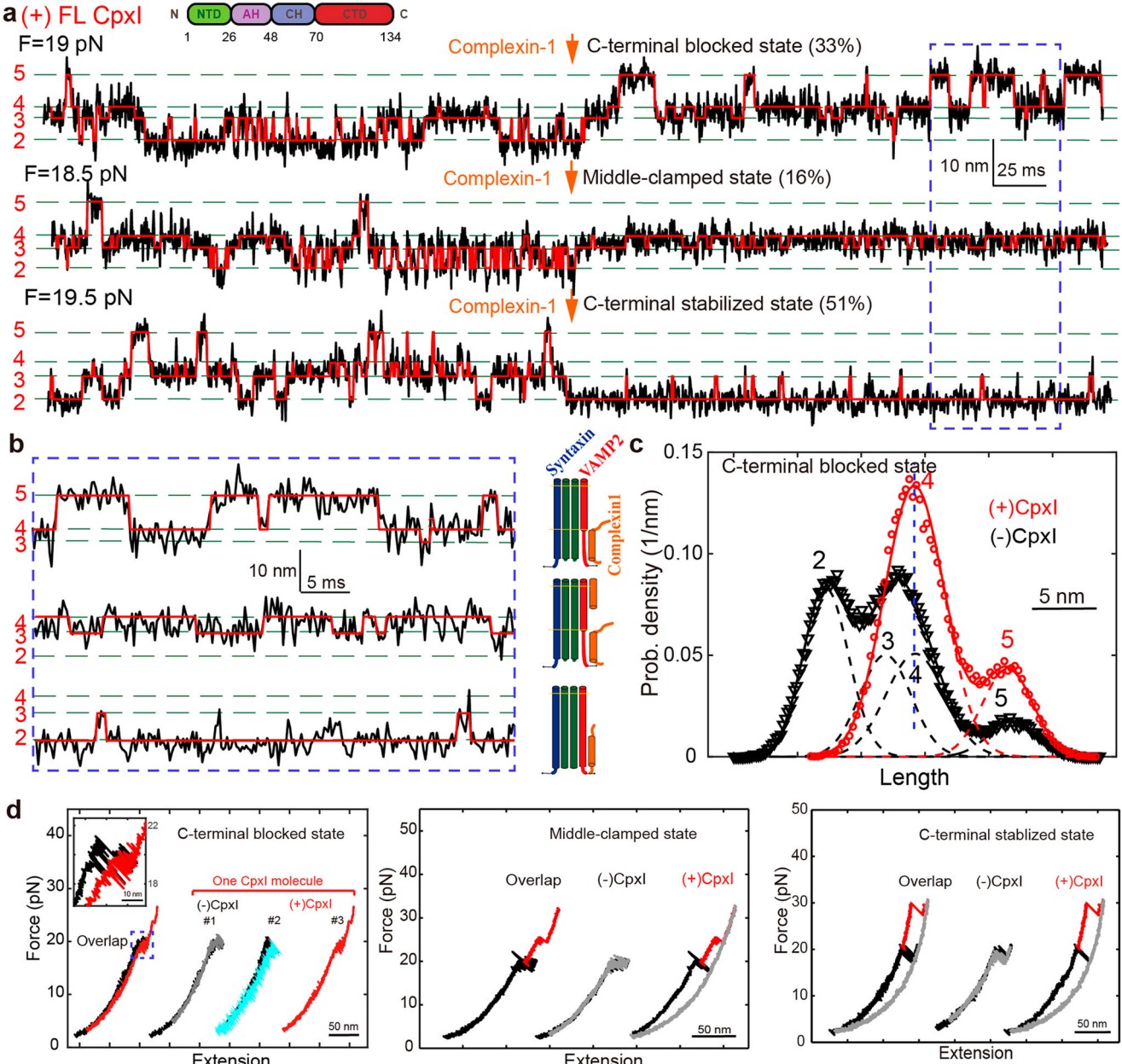

**Fig. 2 CpxI stabilizes partially folded SNARE complex. a** Extension-time trajectories of single SNARE complexes under constant trap separation in the presence of CpxI. The arrow indicates the administration of 8 μM full-length CpxI molecules. The ideal state transitions derived from hidden-Markov model (HMM) are overlaid in red lines. The positions of different states are marked by green dashed lines and labeled with the state numbers. Data were filtered using moving average filter with a time window of 0.2 ms. **b** Close-up views of the region marked by dashed blue rectangles in a. SNARE configurations correspond to the states, black numbers and L (−6 L, +2 L) indicate different layers. **c** Probability distributions of the extensions correspond to the traces in **a** in the presence (red circle) and the absence (black triangle) of CpxI and their best fits by a sum of four Gaussian functions (red and black lines). The open distances for those states in Fig. 2c are 0 nm, 4.6 ± 2.0 nm, 6.9 ± 2.3 nm, 14.6 ± 2.0 nm, respectively, with respect to State 2 (i.e., the open distance for State 2 is assumed to be 0). **d** FECs of a single SNARE complex before (black) and after (red) the addition of 8 μM CpxI, and cyan line is the relaxing FEC after the addition. The FECs of the different cycles generally overlap but were shifted along x-axis for clarity. The inset shows a close-up view of the region in dashed blue rectangle.

crystal structure of Cpx/SNARE complex and mutagenesis experiments have revealed that the CH is directly associated with SNAREs, which is essential for Cpx to execute function[20,21].

In the presence of 8 μM full-length CpxI, 49 SNARE complexes changed their state after the addition of CpxI. For 45% of them (22 molecules), the distribution of SNARE complexes changed from 2~5 states to 3~5 states (Fig. 2a–d, Supplementary Figure 7), which implies that the reversible assembly and disassembly is limited to N-terminal, and the transition of C-terminal is not

allowed (C-terminal-blocked state). As mentioned in supplementary information 6, the signal rate is defined as the ratio of a particular signal to the sum of all signals (Supporting information 12). Moreover, the FEC curves changed largely compared with those in the absence of CpxI (Fig. 2d). Most SNARE complex couldn't zip completely even the force dropped below 5 pN (Fig. 2d, C-terminal clamped state, cyan line), suggesting that CpxI might insert into the C-terminal of SNARE and inhibit the complete zippering of SNARE complex (Fig. 2a–d). HMM

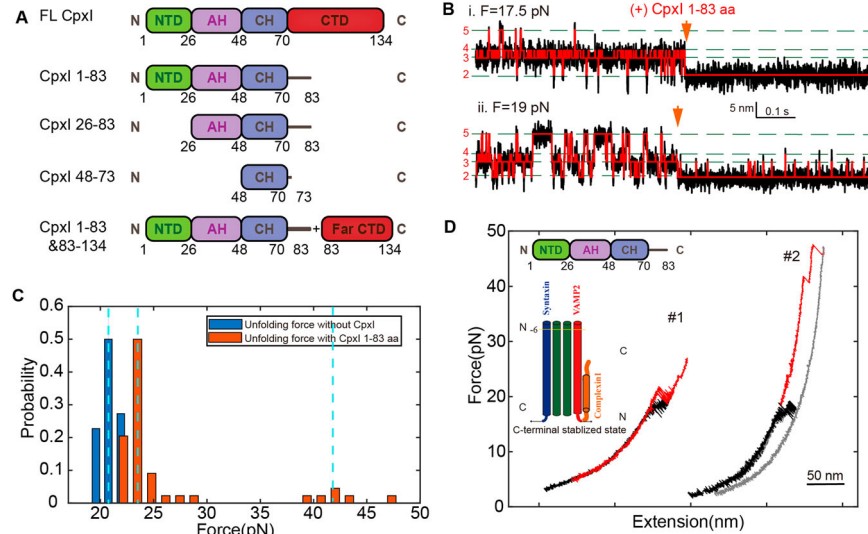

**Fig. 3 Fragment 1-83aa of CpxI stabilizes the SNARE complex at the C-terminal stabilized state. a** Schematics of the domain fragments of the CpxI used in the experiments. **b** Extension-time trajectories of single SNARE complexes under constant trap separation showing SNARE unfolding kinetics before and after the addition of 8 µM fragment 1-83aa of CpxI in presence of an average force of (i) 17.5 pN, and (ii) 19 pN. **c** The distribution of the unfolding force of SNARE complexes with (orange) or without (blue) 1-83 aa fragments. The hopping takes place at ~21 pN in the absence of CpxI, while the average force increases to ~24 pN and some events happen at even higher force of ~43 pN. **d** Force-extension curves (FECs) of a single SNARE complex before (black) and after (red) the addition of 8 µM 1-83 aa. The 1-83 aa domain of CpxI stabilizes the SNARE assembly at the C-terminal stabilized state, for which 59% of the FECs changed slightly with an increase of up to 5 pN (#1) on the unzipped force of SNARE C-terminus, and 34% of the force-extension curve (FEC) changed dramatically with an increase of more than 5 pN (#2) on the unzipped force of SNARE C-terminus.

analysis of extension-time trajectories suggests that the SNARE complex was locked to the C-terminal blocked state in the presence of 8 µM full length CpxI (1-134 aa) (Fig. 2c, before: black, state 2-5; after: red, state 3-5). This signal corresponds to CpxI's clamp function: inhibit the complete assembly of SNARE complex.

We also observed that SNARE complex could be clamped by CpxI into the exactly half-zippered state (middle-clamped state, 7 molecules, 14% signal rate, Figs. 2a, b, d). Accordingly, the dynamic folding of SNARE complex only took place between 3~4 states at this situation (Supplementary Figure 8a). We continued to stretch this complex by ramping up the force, and found that a higher force is essential to break this half-zippered state. This corroborated that CpxI can inhibit the C-terminal assembly of SNARE complexes, and simultaneously stabilize the N-terminal. Such observation also corroborated the ensemble studies on the CpxI's clamp function in physiological state.

Surprisingly, the SNARE complex can also maintain in C-terminal stabilized state after the addition of CpxI (C-terminal stabilized state, 20 molecules, 41% signal rate, Fig. 2a, b, d). Accordingly, SNARE complexes have changed from hopping among 2-5 states to being maintained only in the state 2 with the addition of CpxI (Supplementary Figure 8b). Further stretching of the molecules with a high force would break this C-terminal stabilized state, indicating the ability of CpxI to stabilize the four-helix bundle of SNARE complexes.

Moreover, in the process of consecutive rounds when pulling and relaxing the same SNARE complex, the possibility of the same signal appearing in the subsequent cycle after a certain signal is higher than 80%, indicating that the same CpxI combines with the same SNARE in multiple cycles of pulling and relax (Fig. 2d). This corroborates that the binding of the Cplx on SNARE complex is much stronger than the unfolding of SNARE complex.

**CpxI 1-83aa stabilized the C-terminal of SNARE complex.** To figure out the specific function of multiple domains in the CpxI-

dependent SNARE transition dynamics, we introduced different truncations of CpxI to the single transitioned SNARE complex, including 1-83 aa, 26-83 aa, 48-73 aa, mixture of 1-83 aa and 83-134 aa (Fig. 3a). Remarkably, we found that the signals of SNARE folding in presence of 1-83 aa (lack the CTD of CpxI compared to full-length CpxI) were the most consistent among all kinds of CpxI fragments. In the presence of 8 µM 1-83 aa of CpxI, 49 dynamic SNARE complex molecules changed their state after the addition of CpxI, while another 13 molecules showed no change after the addition. Interestingly, among these 49 SNARE complexes with significant configuration change, 94% (46 molecules) of their extension-time traces suggest that the SNARE complex was stabilized at C-terminal stabilized state, and higher force was required to further dissociate the CpxI molecule (Fig. 3b). And other 3 SNARE complexes (6%) were limited into N-terminal transition. Obviously, the 1-83 aa domain of CpxI stabilizes the SNARE at C-terminal stabilized state. In other words, 1-83 aa can only implement the facilitate function.

The exocytosis of neurotransmitters and hormones is a tightly controlled process that has evolved to meet temporal precision and speed of intercellular communication. Cpx is likely the most controversially discussed SNARE-interacting proteins involved in exocytosis. The study of the regulatory mechanism of SNARE complex has important sense to complete the vesicle fusion theory, and guides the clinical treatment of the related diseases. The exocytosis could be triggered by interaction of multiple domains with the SNARE complex. To examine the effect of CpxI on SNARE zippering, we observed a series of prominent long-dwelling states in the process of fast SNARE transition after the introduction of 8 µM full-length and different kinds of truncated CpxI.

The unfolding forces of the C-terminal in SNARE complexes are normally distributed around an average of ~ 21 pN (Fig. 3c, first blue dashed line, Supplementary Figure 9). However, in the presence of CpxI, the unfolding forces of the C-terminal in SNARE complexes increased differently. In these C-terminal

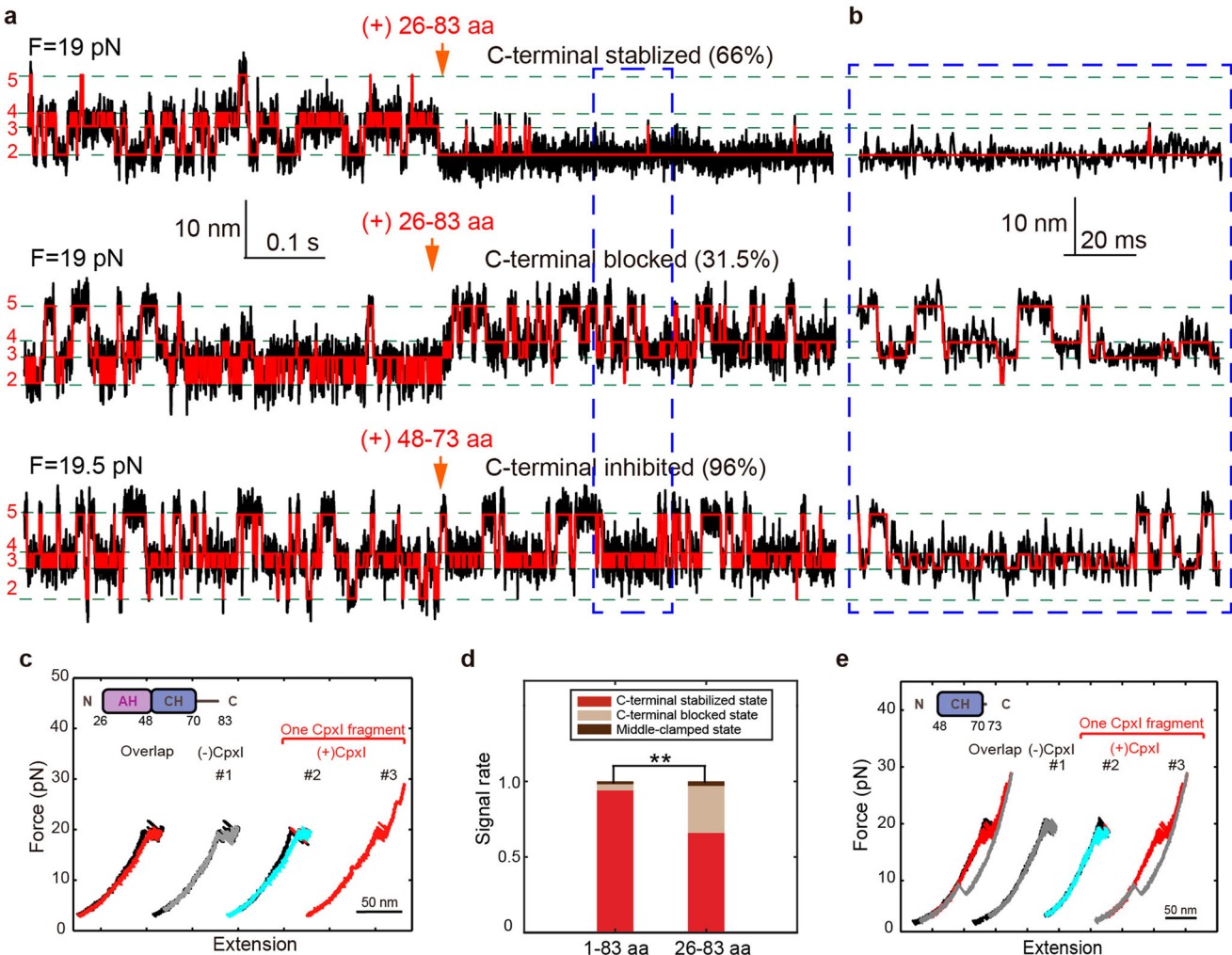

**Fig. 4 The stable function of CpxI NTD and Cpx CH domain can slightly inhibit the assembly of C-terminal of SNARE complex.** In the presence of 8 μM AH-CH domain (26-83 aa), 38 of 66 (57.5%) dynamic molecules changed their state after the addition of the CpxI fragment. **a** Extension-time trajectories and **b** zoom-in view of the traces in the dashed rectangle suggest that 66% of them was stabilized to C-terminal stabilized state after the addition of 8 μM AH-CH domain (26-83 aa) in real time, 31.5% of SNARE complex were locked at the C-terminal blocked state after (red) the addition of 8 μM AH-CH domain. For the addition of a shorter fragment of CH domain (8 μM 48-73 aa), 26 of 32 (81%) dynamic molecules changed their state in the presence of CpxI fragment. **c** FEC curve, black: pulling before the addition of CpxI, gray: relaxing before the addition of Cpx, red: pulling after the addition of CpxI, cyan: relaxing after the addition of CpxI. **d** Summary of the distribution of the different kind of signals introduced by fragments of 1-83 aa or 26-83 aa. **e** FECs of single SNARE complexes in the presence of merely CH domain. 96% of the SNARE complex were locked to the C-terminal blocked state after (red) the addition of CpxI CH domain. The paired t-test was applied to evaluate the statistics with a p-value of *p < 0.04, **p < 0.002, and ***p < 10^-8.

stabilized signals, the unzipping force of the C-terminal for 34% SNARE complexes under study increased by more than 5 pN, and the unzipping force for 59% of the C-terminal SNARE increased up to 5pN (Fig. 3c, d).

**CpxI NTD stabilizes the four-helix bundle of SNARE complex.** The NTD (1-26 aa) is an important domain of CpxI to stabilize the C-terminal of SNARE complex[22,23], specifically ability of CpxI to stabilize SNARE complex may mostly depend on its NTD. To pinpoint the NTD role in the CpxI -dependent SNARE disassembly, we removed the NTD in the CpxI construct. In the presence of CpxI 26-83 aa, 38 dynamic molecules changed their state after the addition of CpxI (Fig. 4a, b). Interestingly, the extension-time traces of 25 molecules (66%) were still stabilized at the C-terminal (Supplementary Figure 10), while 12 SNARE complexes' C-terminal transition were blocked (31.5%, Fig. 4a, b). As for the latter signal, most SNARE complex also couldn't achieve complete zippering even the force dropped below 5 pN

(Fig. 4c, cyan line). The rate of C-terminal stabilized state decreased dramatically, showing that the removal of NTD caused large changes in the CpxI-dependent SNARE disassembly (Fig. 4d), which indicates that the CpxI stabilizes the SNARE complex critically depending on its N-terminal domain (NTD, 1-26 aa). This provides single molecule evidence to support previous reports on that the NTD domain of CpxI localizes to the point where trans-SNARE complex insert into the fusing membranes[24], or lock the C-terminal of SNARE complex in the absence of trigger signal of member fusion[23,25].

The CH domain (48-73 aa) of Cpx is believed to be the smallest fragment that directly binds on SNARE complex[24]. We supplied 8 μM CH domain of CpxI to the single-molecule SNARE complex experiment, and found that the CpxI CH slightly inhibits the assembly of C-terminal of SNARE complex (Fig. 4a, b). Specifically, 26 molecules changed their state after the addition of CpxI, and 96% of their C-terminal transition were slightly blocked (25 molecules, Fig. 4e). However, these molecules showed full assembly as force decreases (the cyan line in Fig. 4e), quite

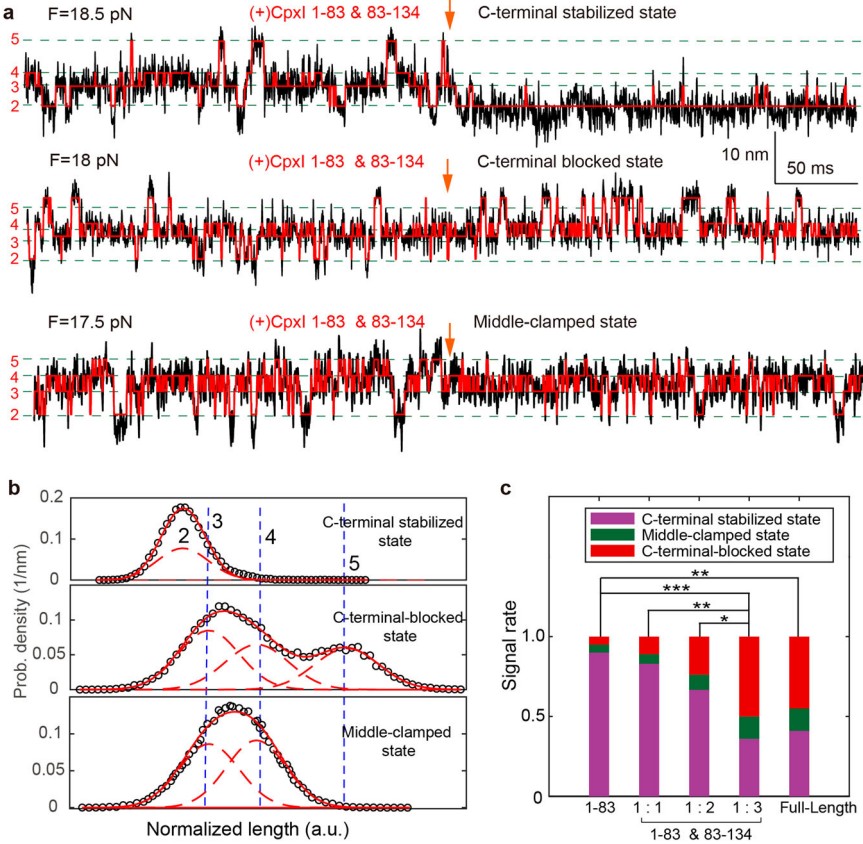

**Fig. 5 CpxI CTD can inhibit the assemble of C-terminal of SNARE complex and stabilize the N-terminal of SNARE complex. a** Extension-time trajectories of single SNARE complexes show SNARE unfolding kinetics before and after the addition of 8 µM fragments of 1-83 aa and 8/16/24 µM fragments of 83-134 aa in real time. **b** Characteristic probability density distributions of the extensions corresponding to the traces in A and their best fits to a sum of four Gaussian functions (red lines). **c** The distribution of different kind of signals introduced by fragment 1-83 aa of Cpx, full-length Cpx, and mixture of 1-83 aa and 83-134 aa with various ratios. With the increase of CTD's concentration, the proportion of C-terminal-blocked state and middle-clamped state increases, meanwhile the proportion of the C-terminal stabilized state decreases. When the ratio reaches 1:3, the types of signals and the signal portion are identical to those for the full-length CpxI. The paired t-test was applied to evaluate the statistics with a *p*-value of *$p < 0.04$, **$p < 0.002$, and ***$p < 10^{-8}$.

different from the effect of other longer CpxI fragments (the cyan line in Fig. 2f, 4c). A much shorter fragment of CpxI 48-73 aa only slightly inhibits the SNARE complex to fold at the CTD and is insufficient to strongly clamp the SNARE complex into the C-terminal blocked state.

A weak interaction occurred between individual CH (48-70 aa) and SNARE complex, since the proper and functional orientation needs the cooperation of other domains in CpxI. The crystal structures of Cpx: SNARE complex show different interaction surfaces for Cpx of different lengths. For Cpx 32-72 aa[26] or 24-73 aa[26], the Cpx CH (48-70 aa) binds to the SNARE complex in the middle domain at around 0 ~ +1 layer; for a shorter Cpx 49-76 aa (containing almost only CH)[21], the Cpx CH binds to the proximal C-terminal of the SNARE complex. When the Cpx peptide is too small, the binding position of CH moves towards the C-terminal of the SNARE complex. This suggests that if the Cpx fragment is too short (only CH), weak or even erroneous interactions are likely to occur, because the functional binding of Cpx requires the coordination of other Cpx domains. Our single-molecule optical tweezers experiment has corroborated that the region with 26–83 aa in CpxI is the 'minimal clamping domain' of the protein[27,28]. Specifically, the CpxI critically depends on its NTD (1-26 aa) to stabilize the SNARE complex.

**CpxI CTD inhibits the C-terminal refolding and stabilizes the N-terminal folding of SNARE complex.** Recent function

analyses with C. elegans Cpx revealed that CpxI inhibitory effects is important but not sufficient for membrane binding[29,30], making it mysterious for the interactions of the Cpx C-terminus for arresting vesicle fusion. Hereby, we elucidate whether CTD is functional in the absence of phospholipid. CpxI CTD can't directly bind to SNARE in the absence of CH, and CTD fragment (83-134 aa) doesn't bind to 1-83 aa (Supplementary Figure 11). We introduce CTD fragment (83-134 aa) combined with 8 µM 1-83 aa of CpxI to interact with the SNARE complex during dynamic assembly. This scheme is similar to the addition of full length CpxI, but each CpxI molecule is separated into two pieces. Unexpectedly, 18 SNARE complex molecules changed their states after the addition of 8 µM 1-83 aa fragment and 8 µM 83-134 aa (with molar ratio of 1:1); 21 SNARE complex molecules changed their state after the addition of 8 µM 1-83 aa and 16 µM 83-134 aa (molar ratio 1:2); 22 SNARE complex molecules changed their state after the addition of 8 µM 1-83 aa and 24 µM 83-134 aa (molar ratio 1:3) (Fig. 5a–c, Supplementary Figure 12). Different combinations of CpxI fragments lead to different types of signal distribution (Fig. 5c).

Interestingly, the proportion of C-terminal-blocked state and middle-clamped state increases with the ramping up of concentration of CpxI CTD fragment, meanwhile the proportion of the C-terminal stabilized state decreases (Fig. 5c). Meanwhile, the probability of the C-terminal-blocked state and middle-clamped state (clamp) also increases with the concentration.

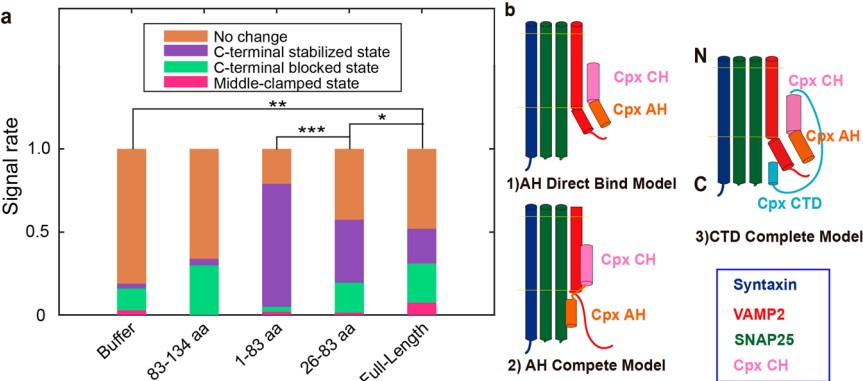

**Fig. 6 Function of the CpxI fragments on the dynamic zippering of SNARE complexes. a** Summary of the signal rate of SNARE complex in presence of various truncations. With the inclusion of the CTD (compare column 5 to column 3), the total proportion of C-terminal-blocked state (green part) and middle-clamped state (pink part) increases, and meanwhile with the remove of NTD (compare column 4 to column 3), the proportion of the C-terminal stabilized state (purple part) decreases. The paired t-test was applied to evaluate the statistics with a p-value of *$p < 0.02$, **$p < 0.03$, and ***$p < 0.0001$. **b** Models of complexin's clamp function. (1) AH direct-binding model. (2) AH compete model. Our existed data prefer the (3)CTD complete model.

Remarkably, more SNARE complexes were clamped to the middle-clamped state: they can't reassemble to a fully assembled complex, and need higher force to break the half-zippered state. Furthermore, the clamping function of Cpx can be reconstituted by fragment 1-83 aa of CpxI and its CTD as separate fragments in vitro (Fig. 5c, last two columns). Thus, physical continuity through the length of CpxI is not required to establish clamp function. CTD of CpxI inhibits the full zipper of SNARE complex, and plays an important role in the clamp function. Moreover, CTD (70-134 aa) is mainly related to clamp function, and separate 1-83aa and CpxI CTD can efficiently reconstitute the inhibitory signal identical to that the full-length CpxI functions (Fig. 5). The Cpx C-terminus competes with SNAP25-SN1 for binding to the SNARE complex and thereby halts progressive SNARE complex formation before the triggering[31]. Our study revealed that the rescue of the function of full-length Cpx requires a high concentration of CpxI CTD, and abundant CTD peptide is able to enhance the inhibitory function of CpxI[31].

## Discussion

CpxI clamps the pre-assembled SNARE complexes into "linker-open state" revealed by magnetic tweezers experiment[32]. However, the SNARE complex is not in a functional state when the CpxI is pre-assembled with the SNARE complex. The unzipping of pre-assembled SNARE complexes occurred at higher force level by ~2 pN on average with the inclusion of CpxI[32]. In contrast, single molecule optical tweezers experiment allows the study of the folding dynamics of SNARE complex in a functional state. Our single molecule optical tweezers experiment suggests that the CpxI can introduce the "C-terminal stabilized state", "middle-clamped state" and "C-terminal blocked state" (Figs. 1, 4 and 5). To determine the critical domain for each type of signal, the signal occurrences for the interaction of SNARE with three representative fragments in CpxI, i.e., 1-83 aa, 26-83 aa, 83-134 aa, is summarized in Fig. 6a.

Moreover, the mechanical tension on single pre-assembled SNARE complexes manifested only in a narrow range, namely between 13 and 16 pN. However, our single-molecule experiment suggests that the interplay of CpxI exists among a broader force range of 5~40 pN. The CH in CpxI can also stabilize the four-helix bundle of pre-assembled SNARE complex, however, our dynamic study using the functional SNARE indicates that the CpxI CH could neither stabilize the SNARE complex nor even

functionally bind to SNARE complex. Our observation is consistent with the in vitro analyses in Hela cells by Rothman and colleagues who demarcated a region comprising amino acids 26–83 of CpxI as the 'minimal clamping domain' of the protein.

The magnetic tweezers study suggests that the deletion of the C-terminal domain had no appreciable effect on any of the aspects of Cpx function, but our study provides direct proof of CpxI CTD's active role in clamp function to inhibit membrane fusion (Fig. 6a).

In the presence of full-length CpxI, SNARE complex can be introduced to three kinds of signals. For C-terminal blocked state, the distribution of SNARE complexes changed from 2~5 states to 3~5 states (Fig. 2a–d,), which implies that the reversible assembly and disassembly is limited to N-terminal, and the transition of C-terminal is not allowed (C-terminal-blocked state). For middle-clamped state, the dynamic folding of SNARE complex only took place between 3~4 states at this situation (Fig. 2a, b, d,). The SNARE complex can also maintain in C-terminal stabilized state after the addition of CpxI (C-terminal stabilized state, Fig. 2a, b, d). Accordingly, SNARE complexes have changed from hopping among 2-5 states to being maintained only in the state 2 with the addition of CpxI.

The 1-83 aa of CpxI can powerfully and dominantly stabilize the four helix-bundle of SNARE complex at the C-terminal (Figs. 3, 1-83 aa in Fig. 6a). The probability for C-terminal blocked state increases once the NTD (1-26 aa) is removed (26-83 aa in Fig. 6a) accompanied by an inhibition of the C-terminal stabilized state. The CTD (83-134 aa) of CpxI, however, dominantly blocks the C-terminal of SNARE (83-134 aa, in Fig. 6a). 1-83 aa of CpxI is insufficient to block the C-terminal zippering of SNARE complex compared with full-length CpxI, for the lack of CTD.

The binding between CH and SNARE is the prerequisite of the clamp function, a bunch of models were proposed to explain the interaction between AH and the half-zippered SNARE[33,34]. Our single molecule experiment on the interaction of various fragment of CpxI and SNARE complex provides single molecule support for the AH binding model (Fig. 6b1) rather than the AH competition model with VAMP c-terminal (Fig. 6b2). This is also consistent with the co-crystallization experiment, which suggests Cpx binds to the C-terminal of VAMP rather than compete[33].

The controversy between these two models (Fig. 6b1, 2) is on the binding site of CpxI AH. Experiment with full length suggest that the SNARE C-terminal can be stabilized/blocked (clamp) after the addition of CpxI, and CpxI fragments 1-83 aa can only

stabilize the C-terminal of SNARE complex. If the full length CpxI realized the clamp function (C-terminal blocked state) by competing model (Fig. 6b2), the binding site by CpxI AH is on t-SNARE without VAMP C-terminal (Vc). When Vc is zippered back to the four-helix bundle, there is no binding site for CpxI AH in the C-terminal stabilized state.

On the contrast, if the full-length CpxI realized the clamp function (C-terminal blocked state) by direct binding model (Fig. 6b1), the binding site by CpxI AH is on Vc. When Vc is zippered back to the four-helix bundle, there is still the binding site for CpxI AH in the C-terminal stabilized state. What's more, for the clamp function, we suggested the CpxI CTD insert the C-terminal of SNARE complex to compete with Vc or t-SNARE (prefer t-SNARE, (CTD complete model in Fig. 6b3) instead of CpxI AH (AH compete model in Fig. 6b2), which is supported by our data mentioned below.

The 1-83 aa of CpxI can powerfully stabilize the four helix-bundle of SNARE complex (Fig. 3). Once the complementary fragment 83-134 aa (mostly part of CTD, without CH) was injected, and 18 of 27 SNARE complexes (67%) showed no change (second column of Fig. 6a, the definition of signal rate in supplementary information 7), which was a negative control to experiments in Fig. 5. The 1-83 aa dominantly stabilizes the SNARE at the C-terminal (1-83 aa in Fig. 6a). The probability for C-terminal blocked state increases once the NTD (1-26 aa) is removed (26-83 aa in Fig. 6a) accompanied by an inhibition of the C-terminal stabilized state. The CTD (83-134 aa) of CpxI, however, dominantly blocks the C-terminal of SNARE (83-134 aa, in Fig. 6a). 1-83 aa of CpxI is insufficient to block the C-terminal zippering of SNARE complex compared with full-length CpxI, for the lack of CTD.

Experiment with mixer of CpxI fragments (83-134 aa and 1-83 aa) suggests the SNARE C-terminal becomes more blocked when the CpxI CTD (83-134 aa) increases, meanwhile the proportion of the C-terminal stabilized state decreases (Fig. 5c). Furthermore, the clamping function of Cpx can be reconstituted by fragment 1-83 aa of CpxI and its CTD as separate fragments in vitro (Fig. 5c, last two columns). Thus, physical continuity through the length of CpxI is not required to establish clamp function. CTD of CpxI inhibits the full zipper of SNARE complex, in addition to CH being responsible for the direct combination of Cpx and SNARE, CTD is responsible for inhibiting the C-terminal assembly of SNARE complexes. Meanwhile, it is quite reasonable for AH to be combined with the C-terminal VAMP which is far away from t-SNARE. Upon the arrival of the stimulus signal facilitating the membrane fusion, this clamp function of CTD is removed, CH still maintains the basic binding, AH can be assembled with VAMP C terminal closer to the SNARE complex, while NTD plays its role of facilitate, and together they stabilize the assembled SNARE complex with four spiral bundles. So that the assembled complex will not be immediately degraded by NSF, until the release of vesicle content is complete.

Our single-molecule observation revealed two functions of the CpxI during interaction with SNARE complex. Full-length CpxI clamps the SNARE complex into half-zippered state without $Ca^{2+}$, while CTD refold back to clamp (CTD could be of helical conformation). Then NTD and Synaptotagmin help SNARE complex complete assembly and stabilize the four-helix bundle of SNARE complex to realize the neurotransmitter release.

In summary, optical tweezers experiment provides direct single-molecule evidence of the interaction between CpxI and SNARE complex in the functional state mimicking the real physiological condition. We built the dual-trap optical tweezers with differential detection and measured the different folding kinetics of SNARE complexes in the absence and presence of both wild-type and mutant/truncated CpxI. A series of synthetic short

fragments of CpxI interplays with the SNARE assembly. In particular, the region 1-83 aa of CpxI can stabilize the four-helix bundle of SNARE motifs (facilitate), and the ability of CpxI to stabilize the SNARE complex critically depends on its NTD. Moreover, separate 1-83aa and C-terminal domains of CpxI can efficiently reconstitute the inhibitory signal of the full-length CpxI. The region with 26–83 aa in CpxI is the 'minimal clamping domain' of the protein[27,28]. The Cpx NTD have critical impact in the facilitate function of Cpx, while Cpx CTD plays an active role in clamp function. Collectively, our results delivered insight into the fundamental mechanisms of exocytosis. Specifically, multiple domains work cooperatively to ensure that the full-length CpxI performs as a calcium-triggered molecular switch - clamp the SNARE complex at half-zippered state at the primed state of vesicle, and accelerates the SNARE complex to complete assembly into fully-folded state in response to (e.g., $Ca^{2+}$-triggered) stimulus in the physiological state. We anticipate that the dual-trap optical tweezers would further be applied to tackle the important disease, such as the Alzheimer's and Huntington's diseases, by tracing the minute misfolded state under mechanical tension.

## Methods

**Purification and labeling of proteins.** The -6 layer cross-linked synaptic SNARE complex consists of VAMP2 (1-92, C2A, Q36C), syntaxin 1 (172-265, C173A, L209C), and SNAP25 (1-206). Substitution or truncation mutations in SNARE proteins and Cpx (C105A, 1-83, 26-83, 26-134, 83-134) were generated by overlap extension polymerase chain reaction (PCR) using respective primers containing the desired non-homologous sequences[35]. All mutations were confirmed by DNA sequence analysis (BioSune, China). Genes corresponding to syntaxin and VAMP2 and the above Cpx were inserted into pET-SUMO vectors through TA-cloning. The proteins were then expressed in E. coli BL21(DE3) cells and purified as described in the manual of ChampionTM pET SUMO Expression System (Invitrogen). Typically, E. coli cell pellets were resuspended in 25 mM HEPES, 400 mM KCl, 10% Glycerol, 10 mM imidazole, pH 7.7 (25 mM HEPES, 800 mM NaCl, 5% Glycerol, 2 mM β-Me, 0.2% Triton X-100, pH 7.5 for Cpx) and broken up by sonication on ice to obtain clear cell lysates. The lysates were then cleared by centrifugation. The SNARE proteins in the supernatant were bound to Ni-NTA resin and washed by increasing imidazole concentrations up to 20 mM. The syntaxin protein was biotinylated in vitro by biotin ligase enzyme (BirA) as described (Avidity, CO). Finally, the His-SUMO tags on both proteins were cleaved directly on Ni-NTA resin by incubating the tagged SNARE proteins/resin slurry with SUMO protease (with a protein-to-protease mass ratio of 100:1) at 4 °C overnight. The SNARE proteins were collected in the flow-through while the His-SUMO tag was retained on the resin. SNAP25 was expressed from the pET-28a vector and purified through its N-terminal His-tag. All SNARE proteins were purified in the presence of 2 mM TCEP to avoid unwanted crosslinking. Peptides of Cpx (48-73 aa) were synthetized by Yochem Biotech. All Cpx was transferred in the HEPES buffer (25 mM of HEPES, 50 mM NaCl, 0.02% CA630, pH 7.4) before optical tweezer experiment.

**Formation and crosslinking of SNARE complex.** Ternary SNARE complexes were formed by mixing syntaxin, SNAP25, and VAMP2 proteins with 3:4:5 molar ratios in 25 mM HEPES, 150 mM NaCl, 2 mM TCEP, pH 7.7 and the mixture was incubated at 4 °C for 30 mins. Here, the TCEP is introduced to avoid formation of polymer through the disulfide bond among multiple SNARE. Formation of the ternary complex was confirmed by SDS polyacrylamide gel electrophoresis. Excessive SNARE monomers or binary complexes were removed from the ternary complex by further purification through Ni-NTA resin using the His-Tag on the SNAP25 molecule. Middle intramolecular crosslinking around the −6 layer between VAMP2_V57C and Syntaxin_L222C occurred at 34 °C, 300 rpm, 16 h at a low concentration of 100 mM in phosphate buffer, 0.5 M NaCl, pH 8.5 without TCEP. Then the 2,260-bp DNA handle containing an activated thiol group at its 5′ end was added to the solution of SNARE complex, which was just concentrated to more than 70 μM, with a SNARE complex to DNA handle molar ratio of 20:1. Intermolecular crosslinking occurred in open air between VAMP2 and the DNA handle, respectively, in 100 mM phosphate buffer, 0.5 M NaCl, pH 8.5. The DNA handle also contains two digoxigenin moieties at the other end. Both the thiol group and Digoxigenin moieties on the handle were introduced in the PCR reaction through primers. The excess of SNARE complexes in the crosslinking mixture was removed after the correct SNARE complex-DNA conjugates were bound to the anti-digoxigenin coated beads.

**Dual-trap optical tweezers with sophisticated control.** The laser source is a continuous wave laser with center wavelength at 1064 nm (J20I-BL-206C, customized with 10 m fiber bundle, Spectra-Physics). The laser was isolated from the

back reflection with an electro-optic isolator (IO-3-1064-VHP, Thorlabs) to prevent damage and destabilization to the laser cavity. A half-wave plate (HW1) in combination with a polarizing beam splitter (PBS1) controls the total power delivered to the dual-trap optical tweezers. A second half-wave plate (HW2) adjusts the power ratio between two laser traps. The first telescope expands the beam into ~4 mm in diameter. The expanded beam was then split and combined by a pair of polarization beam splitters (PBS2 and PBS3). One beam will be kept fixed, while the other beam is steered using a movable mirror driven by a piezostage (Nano-MTA2X, MadCity Labs Inc). A second telescope T2 expands the beam diameter by twice and relayed the piezomirror to the back focal plane (BFP) of the trapping objective (OB1, NA = 1.2, UPLSAPO60XWIR-2, Olympus). Here, in order to maximize the detection efficiency, we adopted two identical objectives (OBJ1 and OBJ2) for trapping and detection. Both objectives are optimized for high transmission (>80%) at the trapping wavelength (1064 nm).

The position detection was performed through two position sensitive detectors (PSDs, DL100-7 PCBA3, Pacific silica) conjugated at the BFP of the detection objective (OB2). To assist for the experiment, a bright-field microscope was co-aligned with the dual-trap optical tweezers. The bright field illumination was provided by a laser emitting diode (LED), and monitored in real-time through a charge-coupled device camera (CCD) (scA640-74fc, Basler). The two laser traps were created by focusing two orthogonally polarized 1064 nm laser beams (Fig. 1a) with a high transmission water-immersion objective. The dual-trap optical tweezers can exert force on a tethered single molecule, specifically, the two microspheres in the trap act as force and displacement sensors. The position traces were detected through two PSDs conjugated to the BFP of the detection objective with back focal plane interferometry[17]. The dual-trap optical tweezers were installed inside an isolated room from environmental noise with well temperature control and air conditioning, specifically, the temperature fluctuation is below 1 °C, while the air flow speed is smaller than 0.1 m/s at the outlet of the air conditioning system. To minimize the noise from human operation and noisy instrument, the laser controller with fan was moved outside of the room with a multimode fiber bundle guiding pump light into the laser cavity, and the instrument is operated, e.g., addition of sample, tether formation, calibration, and measurement, outside the room through a single Labview interface. The instrument operated in high-resolution passive mode, with the location of two traps kept stationary during the recording. Once an increase of the applied force was required, the movable trap separates more from the stationary one through the piezomirror. Despite movable, the beam spot on the back focal plane of the objectives is stationary with intensity distribution depending on the trapped microsphere position. Therefore, the optical tweezers can be calibrated through the power spectrum density of the positional signals,

$$|\tilde{x}(f)|^2 = \frac{k_B T}{c\pi^2\beta[(\frac{\alpha}{2\pi\beta})^2 + f^2]} \quad (1)$$

Here, $\alpha$ is the force constant, typically ranging in 0.01~0.8 pN/nm in our experiment, the conversion constant $c$ is between 0.1~3 mV/nm, and $\beta = 6\pi r\eta$ is a known drag coefficient. Experiments were carried out at room temperature (22 °C) in the HEPES buffer (25 mM of HEPES, 50 mM NaCl, 0.02% CA630, pH 7.4), supplemented with oxygen scavenging system[36] (Supplementary Table 2). The first anti-digoxigenin antibody-coated polystyrene bead (diameter 2.12 μm) suspension was mixed with an aliquot of the mixture (crosslinked DNA handle and the SNARE complex), and the first bead is held in the right optical trap; the second bead (diameter 1.76 μm) coated with Streptavidin, was subsequently captured in another optical trap and brought close to the first bead to form the protein-DNA tether between two beads. Data were acquired at 20 kHz, mean-filtered in situ to 10 kHz, and saved on a computer disk for off-line analysis. In the pulling-relaxation experiment, single SNARE complexes were pulled (relaxed) by increasing (decreasing) the trap separation at a constant speed of 10 nm/s. For dynamics measurement, the separation of the traps was kept constant to record the temporal traces with stochastic transition.

**Trap Geometry**. The extension of the protein-DNA handle tether, $X$, is the sum of the extensions of the DNA handle, $x_{DNA}$, the unfolded polypeptide portion of SNARE complex, $x_p$, and the core structure, h, i.e.,

$$X = x_{DNA} + x_p + h, \quad (2)$$

where h is assumed as the spatial length of the folded portion (such as coiled coil) projected along pulling direction, which also makes contribution to the final extension. In the fully folded state (native state), $h_0 = 2$nm is determined from the x-ray structure of the protein[20], whereas for the fully unfolded state, $h = 0$. And when pulled in an axial direction, $h = 0.15 \times N_s$ nm, where $N_s$ is the number of amino acids in the folded portion and is the helical rise of an amino acid along the helix axis.

According to the experiment, the stretching force $F$ and protein-DNA handle tether $X$ between the inner faces of two polystyrene beads follows,

$$D = X + \frac{F}{k_{trap}} + R_1 + R_2, \quad (3)$$

where $k_{trap} = \frac{k_1 k_2}{k_1 + k_2}$ is the effective trap stiffness, $R_1$ and $R_2$ are the radii of two trapped polystyrene beads and $D$ is the trap separation. The stretching force $F$, the extension $X$, the beads radii $R_1$ and $R_2$, the trap stiffness $k_1$ and $k_2$, the DNA contour length, $L_{DNA} = 768$nm, and the trap separation $D$ can be obtained from the experiment.

**FEC and dynamics fitting**. To characterize the change in contour length of the SNARE-Cpx complex, unfolding and refolding traces were fitted to the worm-like chain model. For the DNA handle, the extension $x_{DNA}$ as a function of the stretching force $F$ is described by the Marko-Siggia model[37,38],

$$F = \frac{k_B T}{P_{DNA}}\left[\frac{1}{4\left(1 - \frac{x_{DNA}}{L_{DNA}}\right)^2} + \frac{x_{DNA}}{L_{DNA}} - \frac{1}{4}\right], \quad (4)$$

where $P_{DNA}$ and $L_{DNA}$ are the persistent and contour lengths of dsDNA, respectively, and $k_B$ is the Boltzmann constant. Similarly, the force for the unfolded polypeptide portion of SNARE complex can be formulated as

$$F = \frac{k_B T}{P_P}\left[\frac{1}{4\left(1 - \frac{x_p}{l_p}\right)^2} + \frac{x_p}{l_p} - \frac{1}{4}\right], \quad (5)$$

where F and $x_p$ are the tension and extension of unfolded polypeptide respectively. The contour length of unfolded polypeptide $l_p$ is related to the number $N$ of amino acid in the polypeptide, which is described as $l_p = 0.4 \times N$, where 0.4 nm is the crystallographic contour length per amino acid[39,40]. And $P_P$ is the persistence length of unfolded polypeptide. $k_B T = 4.1\,pN \cdot nm$ is the energy unit at room temperature.

To characterize the folding pathway of protein, the extension of the SNARE complex in each state is calculated by the model. For each state, the extension consists of two parts: the extension of unfolded polypeptide $l_{ir}$ ($i = 1, 2, 3,$ or 4), which can be calculated by the Marko-Siggia formula and the extension of the folded portion. As described above, except for $l_p$, $P_{DNA}$ and $P_P$, all the other parameters in Eqs. (1)–(4) can be obtained experimentally or theoretically. Therefore, we could fit the experimental extension against the force range of each region (or each state) in FEC, to obtain their best-fit values. The fitting is performed after the FECs are mean-filtered using a time window of 11 ms. The persistence length $P_{DNA}$ and $P_P$ are different more or less for each molecule, so we firstly confirm these two parameters before each FEC fitting of each sample. The FEC region of fully unfolded state is firstly fitted with $P_{DNA}$ (10–50 nm) and $P_P$ (0.5-0.9 nm) as fitting parameters, while the known contour length of fully unfolded protein is taken as fixed parameter. Then the best-fit $P_{DNA}$ and $P_P$ values are utilized as fixed parameters in other FEC regions or states, in which the contour length $l_P$ is to be fitted. Therefore, given the parameters of persistence length of DNA and protein, the structure information of proteins can be derived from fitted contour length of unfolded polypeptide according to Eqs. (2)–(5).

The HMM analysis was normally performed on the whole extension-time trajectory (typically lasted 1–200 s), after the trajectory was mean-filtered to 5 kHz or 1 kHz. We evaluated the number of states by fitting the histogram distribution of the extension with multiple Gaussian functions. Then, the fitting parameters were further optimized in the HMM with a four-state transition model using gradient descent. The HMM analyses yielded the corresponding best-fit parameters from the experimental traces, such as the equilibrium force, and the extension change.

**Statistics and Reproducibility**. The fragments of or complete CpxI interacts with the SNARE complex, and the Extension trace of the SNARE complex may display three different states, i.e., the C-terminal stabilized state (CT), middle clamped state (MC), and the C-terminal Blocked state (CB). Here the probability to each state was defined as the signal rate,

$$\gamma_i = \frac{N_i}{N_{CT} + N_{MC} + N_{CB}} \quad (6)$$

Here, $i = CT, MC, CB$, and CT, MC, CB, represent C-Terminal stabilized, Middle-Clamped, C-terminal Blocked states, respectively. This definition has been used to process the signal rate in Fig. 4d, and Fig. 5c in the main text. Especially, there was only C-terminal inhibited state for CpxI 48-73aa in Fig. 5a.

For we injected only buffer (first column in Fig.6a) though protein channel as control, and the majority (81% of 31 molecules under test) showed no change after the injection. And once the fragment 83-134 aa (mostly part of CTD, without CH) was injected, and 18 of 27 SNARE complexes showed no change (second column of Fig. 6a), which was a negative control of experiments in Fig. 5. In Fig. 6a, we

defined the signal rate as,

$$\gamma_i = \frac{N_i}{N_{Ct} + N_{MC} + N_{CB} + N_{NC}} \qquad (7)$$

$i = CT, MC, CB, NC$. Here, CT, MC, CB, NC, represent C-Terminal stabilized, Middle-Clamped, C-terminal Blocked states, and No Change molecules, respectively.

The paired t-test was applied to evaluate the statistics with a p-value of $*p < 0.04$, $**p < 0.002$, and $***p < 10^{-8}$. The sample sizes were more than n = 30 biologically independent samples.

**Supporting Information**. Supporting Information is available from the publisher's website.

**Reporting summary**. Further information on research design is available in the Nature Portfolio Reporting Summary linked to this article.

## Data availability
The data that support the findings of this study are available from the corresponding author upon reasonable request.

## Code availability
The custom LabVIEW and MATLAB codes used for data acquisition and analysis are available from the corresponding author upon reasonable request.

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

## Acknowledgements
We thank the National Center for Protein Science in Shanghai for use of the instrument. T. H. acknowledges Dr. Y. Gao for help on the SNARE assembly, and single molecule experiment. This work was supported by the National Natural Science Foundation of China (31571346, 31771432). Y.R. acknowledges support from Shanghai and Fudan University.

## Author contributions
T. H., N. F., and F. G. purified the protein, and collected the single-molecule data; T. H., N. F., Y.Y., and Y. R. analyzed the data; Y. R., F. G. and Y. Y. built the instrument; J. L. provided overall guidance and support; T. H. and Y. R. drafted the manuscript, and all authors revised the manuscript.

## Competing interests
The authors declare no competing interests.
