## [Peer Review File · Communications Biology]

Reviewers' comments:

Reviewer #1 (Remarks to the Author):

In this well-presented study, the authors investigate the role of CpxI molecule in the dynamic assembly of SNARE complex under mechanical tension applied through dual-trap optical tweezers. Indeed, the interaction between CpxI and SNARE complex is of crucial importance in Ca²⁺ triggered exocytosis of neurotransmitters. This reveals the importance of the study in indicating the application of optical tweezers in understanding mechanisms involved in vesicle fusion. CpxI is famous for its dual function as either a promoter or an inhibitor of vesicle fusion. In this study, the authors explore how different domains of CpxI affect the folding/unfolding kinetics of SNARE complex and, in agreement with previous studies, they indicate that CpxI 1-83aa, and in particular N terminal domain, is mainly responsible for the facilitate function, while the C terminal domain is mainly related to the inhibitory function of CpxI. The paper is clear in scope and method, and sufficient evidence has been provided for the conclusions. However, there are some concerns that need to be addressed by the authors before further consideration of this manuscript for publication:

- Regarding Fig. 1c and d, the region in the FEC that shows a hop (between state 2 to 5) is a very important region. How can you be sure that the states obtained from HMM coincide with the state transitions? For example, how do we know that MD transition starts at state 3 and finishes at state 4?
- The role of Ca²⁺ and its interaction with synaptotagmin-1 in orchestrating neurotransmitter release is very well-known. However, in this study no experiment was carried out to investigate the role of Calcium Ion. Thus, it is suggested to tone down the statements regarding the role of Ca²⁺ in the discussion and conclusion.
- It is suggested to add the probability density distribution for linker open state and middle clamp state to Fig. 2. Also, it is suggested to provide error bars for the experimental data in Fig.2c.
- More statistical analysis is required, it is suggested to calculate p-value for the data shown in Fig. 6a. For instance, was the portion of linker open state, Middle clamped state, and C-terminal blocked state significantly different in Full-Length CpxI and other cases?
- Please double check the caption of Fig.2, as no Panel "e" is shown in the figure. Also, please double check caption of Fig. 5a regarding black and red lines.

Reviewer #2 (Remarks to the Author):

The authors performed very elegant and sophisticated single molecular studies, employing laser tweezers to measure the forces that control SNARE zippering. They use this powerful technique to investigate the role of Complexin (Cpx) in the SNARE assembly. The manuscript shows very interesting data on how Cpx changes SNARE folding/unfolding. However, several issues outlined below complicate the interpretation of these experiments and make it difficult to relate the experimental observations to the biological functions of the proteins under the study.

1. The SNARE complexes have been cross-linked with disulfide bonds. It appears that all the data are reported for the cross-linked complexes. For the justification, the author state: "The half-zipped state of the N-terminal cross-linked SNARE complex is too short, so we need built a new SNARE pulling system with sable and long half-zipper state. " However, this modification could drastically change the dynamics of the SNARE assembly. Could the obtained results and driven conclusions be transferred to the complexes without the cross-links? This is unclear. Respectively, it is unclear whether the obtained results could be in any way related to the biological functions of Cpx.

2. It is not clear from the manuscript what mutations were introduced to cross-link each of the layers. The text states that only the layer -6 was cross-linked. However, Supplementary figure 3 shows layers -1, -2, and -3 being cross-linked as well. Finally, the mutations shown in the Supplementary Information (#8) do not seem to be sufficient for cross-linking all the layers marked in the

Supplementary figure 3. The authors need to state very clearly what was cross-linked and to show how the cross-links affected the force-extension graphs. They also need to discuss how the cross-links could affect the data interpretation.

3. The relationship between states and separation distances is not entirely clear. Figure 1c shows the states 2 and 5 having a separation distance of about 30 nm. From figures 1 d and 2 c, this distance appears to be 15 nm. Are these the same measurements? The scales need to be clarified.

4. I am not convinced that the states 3 and 4 are actually separate states. Figure 1 d shows these two states very close to each other. Figure 2 c shows three states without Cpx (2, 3, and 5) and three states with Cpx (2, 4, and 5), and the peaks for the states 3 and 4 match almost exactly. I would interpret this data as only three states. This result would also match an earlier study (Gao et al., cited by the authors), who also discovered three states. If the authors are convinced that they actually discovered the fourth state, they need stronger data to support this claim.

5. The result presented in figure 2 c is very interesting, since it shows that the state 2 disappears in the presence of full length Cpx. It would be very interesting to investigate the state 2 (which disappears after Cpx is added) in more detail and to determine which layers are separated in the state 2. This can be done, for example, by structural molecular modeling. What would be the distance between the attached residues of VAMP2 and Syntaxin for the State 2? What layers should be separated to achieve this distance? How does this agree (or disagree) with the known models for Cpx binding to the SNARE complex?

6. Figure 3 shows the unfolding force for Cpx mutant versus no Cpx. For the comparison, it also needs to show the force for full length Cpx. This data is not reported in earlier figures. The text refers for this to the Supplementary figures 5 A,B. However, the supplementary material has the figure 5 with a single panel which shows examples of force-extension curves but does not show the distribution of the unfolding forces.

7. What is "signal rate" (figs 4d, 5 c, and 6 a), and how is it related to anything? I was not able to find in the manuscript any information on how the "signal rate" was derived.

8. The distributions for the state occupancies (figs 2 c and 5 b) should be reported for every mutant. As is, these distributions are only shown for full length Cpx and for CTD mutant. What happened with all the mutants shown in figures 3 and 4?

9. The model shown in Fig. 6 b should be discussed in relation to other models. The same relates to the results. For example, the 1-83 Cpx fragment drastically promotes SNARE assembly (Fig. 3a). This result seems to contradict a bunch of earlier studies, which showed that Cpx AH promotes the SNARE separation, not assembly. Of course, these direct single molecular studies are very powerful, and they may show that the earlier interpretations need to be reconsidered. But this discrepancy must be discussed.

Minor:

1. Please state clearly what are "linker-open", "C-terminal-blocked", and "C-terminal-inhibited", and "middle-clamped" states" – are they states 2, 5, 4, and 3 ? The table with names, numbers, and separation distances for the states would help.

2. All the interpretations and explanations should be moved to Discussion from Results. For examples, the paragraphs like:

"The optical tweezers were invented by A. Ashkin in 1986, who won the Nobel Prize 74 in Physics in 2018 because of the invention of optical tweezers and the application in 75 biology. The application of optical tweezers spans from colloidal physics, cell biology, to single molecule biophysics."

Or:

"Structural studies can provide an insightful explanation. The crystal structures of Cpx: SNARE complex show different interaction surfaces when Cpx of different lengths is used. For Cpx 32-72 aa (23 or 24-73 aa (23)), the binding position of Cpx CH corresponds to the middle part of the SNARE complex (0 ~ +1 layer); for a shorter Cpx 49-76 aa (containing only CH) 18, the Cpx CH binds to the proximal C-terminal of the SNARE complex. When the Cpx peptide is too small, the binding position of CH moves towards the C-terminal of the SNARE complex. This suggests that if the Cpx fragment is too short (only CH), weak or even erroneous interactions are likely to occur, because the functional binding of Cpx requires the coordination of other Cpx domains. Our observation supports the hypothesis that CpxI AH-CH is the "minimum Clamp unit". These should not be in the Results section.

2. The language needs improvement

3. Figs S1-S3 are not cited in the right order

Reviewer #1 (Remarks to the Author):

- Regarding Fig. 1c and d, the region in the FEC that shows a hop (between state 2 to 5) is a very important region. How can you be sure that the states obtained from HMM coincide with the state transitions? For example, how do we know that MD transition starts at state 3 and finishes at state 4?

Response: Thank you very much for your comments. About the state classification of SNARE complex assembly/disassembly, we first categorized the signal into several states according to the raw data of FEC and Extension-time trace, and then we fitted the FEC curve with worm-like chain (WLC) model for each FEC curve in Fig1.c, and determine the lengths for folded and unfolded length of the protein complex. For instance, the force for the unfolded polypeptide portion of SNARE complex can be formulated as

$$F = \frac{k_B T}{P_p} \left[\frac{1}{4 \left(1 - \frac{x_p}{l_p}\right)^2} + \frac{x_p}{l_p} - \frac{1}{4} \right]$$

where F and x_p are the tension and extension of unfolded polypeptide respectively. The contour length of unfolded polypeptide l_p is related to the number N of amino acid in the polypeptide, which is described as $l_p = 0.4 \times N$, where 0.4 nm is the crystallographic contour length per amino acid. And P_p is the persistence length of unfolded polypeptide. $k_B T = 4.1 \text{ pN} \cdot \text{nm}$ is the energy unit. Note that during fitting, the persistence length and the contour length of each fragment (folded part, unfolded part, and the DNA handle) are chosen for the respective fragments. The single molecule experiment for the SNARE complex can be modeled and understood in the following Fig. R1.

Fig. R1. Schematic of the single molecule experiment. The SNARE core, poly-peptide, and DNA handle are all fit with separate worm-like chain model (with respective persistence lengths), while the dual-trap optical tweezers can be modeled as an equivalent single trap.

The whole extension-time trajectory was mean-filtered to 5 kHz or 1 kHz to evaluate the number of states by fitting the histogram of the extension with multiple Gaussian functions. Combined with the well know crystal structure of the SNARE complex, and the possible binding mechanism of the CpxI molecule, we are able to determine the configuration of the protein complex for each state, and estimate the extension for each state from the extension-time traces.

The estimated extension for each state will be used as initial guess for HMM fitting to more precisely determine the extension for each state. The fitting parameters were further optimized in

the HMM with a four-state transition model using gradient descent. The HMM analyses yielded the corresponding best-fit parameters from the experimental traces, such as the equilibrium force, and the extension change.

To characterize the folding pathway of protein, the extension of the SNARE complex in each state is calculated by the model. For each state, the extension consists of two parts: the extension of unfolded polypeptide l_{ir} ($i=1, 2, 3,$ or 4), which can be calculated by the Marko-Siggia formula and the extension of the folded portion. As described above, except for l_P , P_{DNA} and P_P , all the other parameters can be obtained experimentally or theoretically. Therefore, we could fit the experimental extension against the force range of each region (or each state) in FEC, to obtain their best-fit values.

As for MD, namely transition between 3-4 states, the fitting of these states is also consistent with the literature (Ma et al., 2015). This state had a maximum lifetime of ~ 1 ms and rapidly transitioned between state 3 and state 5. We also confirmed the transition experimentally in FECs and extension traces plotted at 1 kHz bandwidth (Figure R2).

Fig. R2. Top: Extension-time traces, Middle: Force-time traces, and Bottom: trap separation.

As mentioned above, we can calculate the corresponding SNARE complex conformation according to the structural length of each state. In partial zipper state 3, the VAMP2 CTD and T-SNARE CTD unfold from +8 to +3 and +5 layers respectively, while the rest of the SNARE complex are mainly helical as in the quad helical bundles. This structure is consistent with that of different Trans-SNARE complexes on yeast vacuoles (Schwartz and Merz, 2009). In MD expansion state 4, VAMP2 expands further into the central ion layer. Thus, MD consists of an ionic layer (0 layer) and +1 and +2 layers, which contain only leucine and isoleucine residues known to form strongly coiled coils, resulting in MD having higher mechanical stability than CTD, as well as their different transitions (Ma et al., 2015).

2. The role of Ca^{2+} and its interaction with synaptotagmin-1 in orchestrating neurotransmitter

release is very well-known. However, in this study no experiment was carried out to investigate the role of Calcium Ion. Thus, it is suggested to tone down the statements regarding the role of Ca^{2+} in the discussion and conclusion.

Response: We appreciate the reviewer's insightful suggestion. According to suggestion, we have toned down the statements about the Ca^{2+} in the discussion and conclusion. Please see the revisions marked in red in our update.

3. It is suggested to add the probability density distribution for linker open state and middle clamp state to Fig. 2. Also, it is suggested to provide error bars for the experimental data in Fig.2c.

Response: Thank you for your precious comments and advice. We have added the probability density distribution for linker open state (we change the name of this state to C-terminal stabilized state in whole manuscript) and middle clamp state in supporting information 7 (Figure R3 and Figure S6). In addition, each HMM analysis was normally performed on one extension-time trajectory. We evaluated the number of states by fitting the histogram distribution of the extension with multiple Gaussian functions. Then, the fitting parameters were further optimized in the HMM with a four-state transition model using gradient descent. We added a statistical table to represent the range of distribution of each state. Specifically, in Fig. 2c, we included the state position and the standard deviation. For example, the open distances for those states in Fig. 2c are 0 nm, 4.6 ± 2.0 nm, 6.9 ± 2.3 nm, 14.6 ± 2.0 nm, respectively, with respect to State 2 (i.e., the open distance for State 2 is assumed to be 0). All those values are updated in the main manuscript.

Fig. R3. HMM analysis of C-terminal stabilized state and middle clamp state. Probability distributions of the extensions correspond to the traces in Fig 2a (a. middle-clamped state; b. C-terminal stabilized state) in the presence (red circle) and the absence (black triangle) of CpxI and their best fits by a sum of four Gaussian functions (red and black lines).

4. More statistical analysis is required, it is suggested to calculate p-value for the data shown in Fig. 6a. For instance, was the portion of linker open state, Middle clamped state, and C-terminal blocked state significantly different in Full-Length CpxI and other cases?

Response: Thank you for your precious comments and advice. We have applied paired t-test to compare the significance of each group in Fig. 6a, and evaluated the p-value to reject the null hypothesis. All the pairs that do not reject the null hypothesis are labeled in the Fig. 6a, with the p-value described in the Figure caption.

We also performed the t-test for Fig. 4d, and 5c accordingly (Fig R4), and revised the description.

Fig. R4. Figure 4d. Summary of the distribution of the different kind of signals introduced by fragments of 1-83 aa or 26-83 aa. The paired t-test was applied to evaluate the statistics with a p-value of * $p < 0.04$, ** $p < 0.002$, and *** $p < 10^{-8}$. Figure 5c. The distribution of different kind of signals introduced by fragment 1-83aa of Cpx, full-length Cpx, and mixture of 1-83 aa and 83-134 aa with various ratios. The paired t-test was applied to evaluate the statistics with a p-value of * $p < 0.04$, ** $p < 0.002$, and *** $p < 10^{-8}$. Figure 6a. Summary of the signal rate of SNARE complex in presence of various truncations. The paired t-test was applied to evaluate the statistics with a p-value of * $p < 0.02$, ** $p < 0.03$, and *** $p < 0.0001$.

5. Please double check the caption of Fig.2, as no Panel "e" is shown in the figure. Also, please double check caption of Fig. 5a regarding black and red lines.

Response: We thank the reviewer for pointing this out. We apologize for our omission. We have removed the e in the caption of Fig.2, and have rewritten the caption of Fig. 5a.

We have also checked all the numbering and labeling of throughout all the figures.

Reviewer #2 (Remarks to the Author):

The authors performed very elegant and sophisticated single molecular studies, employing laser tweezers to measure the forces that control SNARE zippering. They use this powerful technique to investigate the role of Complexin (Cpx) in the SNARE assembly. The manuscript shows very interesting data on how Cpx changes SNARE folding/unfolding. However, several issues outlined below complicate the interpretation of these experiments and make it difficult to relate the experimental observations to the biological functions of the proteins under the study.

Response: We appreciate that the reviewer thinks our “manuscript shows very interesting data on how Cpx changes SNARE folding/unfolding”. In the update, we have considered all the issues raised by both reviewers and made corresponding revisions. We wish the reviewers will be satisfied with our update.

1. The SNARE complexes have been cross-linked with disulfide bonds. It appears that all the data are reported for the cross-linked complexes. For the justification, the author state: “The half-zipped state of the N-terminal cross-linked SNARE complex is too short, so we need built a new SNARE pulling system with sable and long half-zipper state. “However, this modification could drastically change the dynamics of the SNARE assembly. Could the obtained results and driven conclusions be transferred to the complexes without the cross-links? This is unclear. Respectively, it is unclear whether the obtained results could be in any way related to the biological functions of Cpx.

Response: Thank you for your careful review. (1) As the assembly of SNARE complex is very complex and may subject to miss-folding, most studies on the dynamic assembly of SNARE complex try to reduce the miss-folding(Choi et al., 2016) and verify the correct assembly(Yin et al., 2016). In this study, for each single molecule of SANRE complex, we will first pull and relax a cycle to check whether it display the classical assemble signal, so as to confirm its correct assembly before we record the single molecule traces. In this process, SNARE will be disassembled to state 5. If there is no cross-linking, the SNARE complex will completely and irreversibly depolymerize into monomer, and we won't be able to record the dynamic assembly of the protein complex. At the same time, adding disulfide bonds between proteins without covalent connections is a common and effective technique(Gao et al., 2012; Ma et al., 2015). Also, we compared the pull up curves of SNARE complexes in N-terminal, -6, -2 crosslink (similar to the figure R5 below).

Fig. R5. Single molecule experiment on N-terminal cross-linked SNARE complex. a. Schematic of the single molecule experiment with the protein complex crosslinked at N-terminal. b. FEC of N-terminal cross-linked SNARE complex without Cpx, which show a “hopping+rip” signal at 17-20 pN. c. FEC of N-terminal cross-linked SNARE complex with Cpx.

(2) We have studied the interplay of the Cplx molecule with the N-terminal crosslinked SNARE, the experimental settings are shown in Figure R3a. The original “hoping+rip” dynamic folding signal was inhibited when the force increases in the range of 17 to 20 pN. In other words, it was completely stable in the fully assembled conformation under the original disassembled force, so that higher force and thus more energy is needed to break this assembly. Ensemble study suggests that CpxI binds to and stabilizes the quad helix bundles of SNARE complex. However, ensemble study lacks the recording of a variety of folded intermediate states of SNARE complex, i.e., CpxI can only be combined with fully assembled SNARE complex. In real physiological conditions, the CpxI can also bind to intermediate states of SNARE.

(3) We try to simulate the state of mutual exclusion of the membrane *in vivo* by applying opposite forces to C-terminal of the SNARE complex and making it stay in a half-folded state. This is to explore the interaction between CpxI and partially folded SNARE complexes.

As to whether the crosslinking will affect the real biological function, we here emphasize that the crosslinking is to guarantee the reversible transition of the same SNARE complex so we can record the transition traces in a longer time span. In fact, the fully unfolded state takes place at a very high force range, e.g., >50 pN. In real physiological conditions, SNARE drives the membrane fusion through the electrostatic force. In this force range, the SNARE complex will mostly keep in the folded or partial folded state, therefore, our observation within the force range of 0~30 pN is comparable to how the interaction of CpxI and SNARE works in the real physiological conditions.

In contrast to previous study on the interaction of CpxI with fully folded SNARE, single

molecule optical tweezers allow the investigation of the interaction of CpxI with partially folded SNARE complex, therefore we move one step further towards the understanding of the real function in vivo as the SNARE complex may experience a series of partially folded state when exerting real biological function between membranes.

2. It is not clear from the manuscript what mutations were introduced to cross-link each of the layers. The text states that only the layer -6 was cross-linked. However, Supplementary figure 3 shows layers -1, -2, and -3 being cross-linked as well. Finally, the mutations shown in the Supplementary Information (#8) do not seem to be sufficient for cross-linking all the layers marked in the Supplementary figure 3. The authors need to state very clearly what was cross-linked and to show how the cross-links affected the force-extension graphs. They also need to discuss how the cross-links could affect the data interpretation.

Response: Thank you for your comment. We are very sorry that we neglected to indicate all the cross-linked mutation information, and in the revision, we included the sequences of all the mutants in supplementary materials and also cited in the following. Every kind of cross-link was between the mutant cysteines in Syntaxin and VAMP2 (-1 layer: VAMP C57, Syntaxin C222; -2 layer: VAMP2 C54, Syntaxin C219; -3 layer: VAMP2 C46, Syntaxin C215; -6 layer: VAMP2 C36, Syntaxin C205). We have conducted experiments on all types of cross-linking, among which -2 layer and -6 layer's SNARE complex performance were better. The signal of them were shown in the figure below (figure R6).

-1 layer

VAMP2 (1-92, C2A, V57C)

SAGGMSATAATVPPAAPAGEGGPPAPPNLTSNRRLOQTOAQVDEVVDIMRVNVDK
CLERDQKLSLDDRADALQAGASQFETSAAKLKRKYWWKNGGSGNGSGGLCTPSRGGD
YKDDDDK

Syntaxin (172-265, C2A, L222C)

SAGGGNPAIFASGIIMDSSISKQALSEIETRHSEIIKLENSIRELHDMFMDMAMCVESQ
GEMIDRIEYNVEHAVDYVERAVSDTKKAVKYQSKARRKKGGSGNGGSGSGLNDIFEAQK
IEWHEDYKDDDDK

-2 layer

VAMP2 (1-92, C2A, V54C)

SAGGMSATAATVPPAAPAGEGGPPAPPNLTSNRRLOQTOAQVDEVVDIMRVNCDK
VLERDQKLSLDDRADALOAGASQFETSAAKLKRKYWWKNGGSGNGSGGLCTPSRGGD
YKDDDDK

Syntaxin (172-265, C2A, M219C)

SAGGGNPAIFASGIIMDSSISKQALSEIETRHSEIIKLENSIRELHDMFMDCAMLVESQ
EMIDRIEYNVEHAVDYVERAVSDTKKAVKYQSKARRKKGGSGNGGSGSGLNDIFEAQKIE
WHEDYKDDDDK

-3 layer

VAMP2 (1-92, C2A, M46C)

SAGGMSATAATVPPAAPAGEGGPPAPPNLTSNRRLOQTOAQVDEVVDICRVNVNDKV
LERDQKLSLDDRADALQAGASQFETSAAKLKRKYWWKNGGSGNGSGGLCTPSRGGDY
KDDDDK

Syntaxin (172-265, C2A, M215C)

SAGGGNPAIFASGIIMDSSISKQALSEIETRHSEIIKLENSIRELHDCFMDMAMLVESQ
EMIDRIEYNVEHAVDYVERAVSDTKKAVKYQSKARRKKGGSGNGSGSGGLNDIFEAQKIE
WHEDYKDDDDK

-6 layer

VAMP2 (1-92, Q36C)

SAGGMSATAATVPPAAPAGEGGPPAPPNLTSNRRLQQTCAQVDEVVDIMRVNVDK
VLERDQKLSLDRADALQAGASQFETSAAKLRKYWWKNGGSGNGSGGLCTPSRGGD
YKDDDDK

Syntaxin (172-265, L205C)

SAGGGNPAIFASGIIMDSSISKQALSEIETRHSEIIKCENSIRELHDMFMDMAMLVESQ
GEMIDRIEYNVEHAVDYVERAVSDTKKAVKYQSKARRKKGGSGNGSGSGGLNDIFEAQ
KIEWHEDYKDDDDK

SNAP-25 (1-206, C85S/C88S/C90S/C92S)

MAEDADMRNELEEMQRRADQLADESLESTRMLQLVEESKDAGIRTLVMLDEQGE
QLERIEEGMDQINKDMKEAEKNLTDLGKFSGLSVSPSNKLNKSSDAYKKAWGNNQDGVV
ASQPARVVDEREQMAISGGFIRRVTDARENEMDENLEQVSGIIGNLRHMALDMGNEID
TQNROIDRIMEKADSNKTRIDEANQRATKMLGSG

The sequences with underline were the native sequences, red marked amino acids were mutated amino acids, the FLAG tag (DYKDDDDK) at the C-terminal was used for protein purification, but was not used in this assay. SAGG at N-terminal and GGSGNGSGG at C-terminal were protein linkers, which were designed to increase the flexibility of protein. The sequence 'GLNDIFEAQKIEWHE' marked in blue was designed for biotinylation.

We have tried all the four crosslinking sites (-1, -2, -3, -6), and found that the crosslinking at -1 or -3 layer was unstable, as a result, the efficiency to form a tether is very low, and the tether would easily break under a force of 16 pN. We have successfully formed tether with high efficiency for the crosslinking at -2 and -6 layers. For the crosslinked SNARE at -2 layer, we have successfully collected the dynamic transition signal at ~ 16 pN. In the revision, we take the chance to include part of the signal in the supplementary information (See Fig. S4). In figure R6a, the de-assembly signals obtained by the experiments of the SNARE complex of -2 layer was hopping at 10 nm, corresponding to the de-assembly process of C-terminal and partial N-terminal of 0 to -2 layer (about 28+8 amino acids in total). In figure R6b, the de-assembly signal hops at 20nm, which correspond to the de-assembly process of the C-terminal of the SNARE complex and partial N-terminal of the 0-6 layer (about 28+22 amino acids in total). Because the -2 layer is too close to the 0 layer, and the de-assembly protein fragment covering the N-terminal is too short, insufficient for us to study the interplay of the Cplx with the N-segment of SNARE complexes. Therefore, the subsequent experiments are mainly based on the structure of the -6 layer with longer hopping.

Fig. R6. -2 layer and -6layer cross-linked SNARE complex single molecule experiment. a. FEC of -2layer cross-linked SNARE complex without Cpx. b. FEC of -6 layer cross-linked SNARE complex without Cpx. c. Extension-time trace of -2layer cross-linked SNARE complex with Cpx.

We also recorded the dynamic transition of the -2 layer crosslinked SNARE complex. The -2 layer SNARE complex is stretched to a hopping signal in presence of a fixed trap separation. We observed the transition among three state of SNARE complex in equilibrium (Fig. R6c). Then CpxI was added through protein channels, and significant long stay pauses were observed during the rapid transition from SNARE complexes (in green, Fig. R6c), where the SNARE complexes were stabilized in a quad helical bundle state, revealing a new CpxI dependent state. This is consistent with the signal that the -6layer SNARE complex could be stabilized in the four-helical bundle state (i.e., the C-terminal stabilized state in Fig. 2).

In summary, we have tried to crosslink the SNARE complex at various layers and verified that the -6 layer allows stable performance and maximum interaction region on the SNARE complex with the Cplx molecule. Although different crosslinking site shows slightly different force-extension graph (mainly the open distance is different), the interaction of the Cplx and SNARE complex in the same region suggests consistent mechanism. On the data interpretation, the major principle is the same, the only difference is that when we fit the force-extension curve with the worm-like chain (WLC) model, a different contour length for each of the four-helix bundle, polypeptide according to the crosslinking site was used according to the crosslinking site. In order to maximize the interaction region on the SNARE, most of our experiment was conducted with -6 layer crosslinking on SNARE complex.

3. The relationship between states and separation distances is not entirely clear. Figure 1c shows the states 2 and 5 having a separation distance of about 30 nm. From figures 1 d and 2 c, this distance appears to be 15 nm. Are these the same measurements? The scales need to be clarified.

Response: Thank you for your comment. Fig. 1c shows the Force-Extension-Curve (FEC) during the ramping up of the applied force through increase of the trap separation. It is difficult

to read the extension difference between state 2 and 5 in the overall figure (Fig. 1c), therefore, we zoom in the part of the curve between 2 red dots marked on states 2 and 5. From the zoom-in view (inset of Fig. 1c), the extension change is about 20 nm at a force of ~ 18 pN, when hopping mostly takes place. In Fig. 1d, we basically kept the trap separation fixed and applied a constant average force on the molecule, the average force is ~ 19 pN in Fig. 1d., with a difference in distance of ~ 15 nm between states 2 and 5.

We also include those figure panels here for the reviewers to clearly read. Fig. 1c and 1d are measured on the same SNARE complex. The only difference: (1c) the force is ramping up, (1d) the average force is constant.

Fig. 2c is evaluated on the SNARE in molecule in the presence (red curve) and absence (black curve) of the Cplx molecule. The black curve is comparable with the trace in Fig. 1d, the minute difference is the average force (~ 19 pN in fig. 1d, ~ 19 pN in Fig. 2c). Despite the force shows minute difference the open distance between states 2 and 5 is very close (~ 15 nm in Fig. 1d, ~ 15 nm in Fig. 2c).

Fig. R7. Figure 1c: force-extension curves (FECs) of a single SNARE complex during pulling (black) and relaxing (gray). Figure 1d: extension-time trajectories of single SNARE complexes under constant trap separation. Figure 2c: probability distributions of the extensions correspond to the traces in a in the presence (red circle) and the absence (black triangle) of CpxI and their best fits by a sum of four Gaussian functions (red and black lines). The open distances for those states in Fig. 2c are 0 nm, 4.6 ± 2.0 nm, 6.9 ± 2.3 nm, 14.6 ± 2.0 nm, respectively, with respect to State 2 (i.e., the open distance for State 2 is assumed to be 0).

4. I am not convinced that the states 3 and 4 are actually separate states. Figure 1 d shows these two states very close to each other. Figure 2 c shows three states without Cpx (2, 3, and 5) and three states with Cpx (2, 4, and 5), and the peaks for the states 3 and 4 match almost exactly. I would interpret this data as only three states. This result would also match an earlier study (Gao et al., cited by the authors), who also discovered three states. If the authors are convinced that they actually discovered the fourth state, they need stronger data to support this claim.

Response: Thank you for your careful review. First, we want to clarify that Figure 2 c shows four states without Cpx (2, 3, 4 and 5) and two states with Cpx (4, and 5). First, the difference between the length of each structure and the length of the fully unfolded protein can be used to

calculate how many amino acids have been assembled at the specific time. Therefore, each curve corresponds to a different state of protein folding. The transition between 3-4 states corresponds to the folding/unfolding of the middle domain (MD), and this transition was clearly seen in some FECs and extension traces plotted at 1 kHz bandwidth. Moreover, the transition state is also consistent with the literature.

To point out that only three states (corresponding to States 2, (3, 4), 5 in our manuscript) were reported in reference of (Gao et al, 2013), they crosslink Syntaxin and VAMP2 at the linker domain at the N-terminal of Syntaxin and VAMP, and the fully open state (State 5) was irreversibly generated from State 4 in the form of a large rip (See the comparison in Ma et al., Elife, 2015). We bypass the large rip and crosslink the Syntaxin and VAMP2 at the -6 layer site, and observed consistent dynamics with the report in Ma et al. Elife, 2015. Such choice of the crosslinking site allows the reversible transition among all the states (State 2 through 5) under study.

In conclusion, the states observed in our experiment is consistent with previous report in (Gao, et al, 2013, and Ma et al, 2015). We also clarified this point in the essay at page 9.

5. The result presented in figure 2 c is very interesting, since it shows that the state 2 disappears in the presence of full length Cpx. It would be very interesting to investigate the state 2 (which disappears after Cpx is added) in more detail and to determine which layers are separated in the state 2. This can be done, for example, by structural molecular modeling. What would be the distance between the attached residues of VAMP2 and Syntaxin for the State 2? What layers should be separated to achieve this distance? How does this agree (or disagree) with the known models for Cpx binding to the SNARE complex?

Response: Thank you for your careful review. The FECs of the SNARE complexes lacking the linker domain in VAMP2 has proven that the state 2 is caused by the unfolding of the Linker domain(Gao et al., 2012). State 2 illustrate the state that only the linker domain unfolds (See the following figure for comparison between States 1 and 2 for the difference.

In State 2, only the linker domain on the SNARE complex was unfolded. As a reference, the CTD domain in SNARE complex can be unfolded to the neighboring State 3. The extension difference between State 2 and 3 is around 4.6 nm according to our HMM fitting in Fig 2 (State 2, 727.3 nm, State 3, 731.9 nm), while the crystal structure suggests that the CTD has a length of 33 aa ($32 \text{ aa} / 3.2 \text{ aa/turn} * 0.51 \text{ nm/turn} = 4.5 \text{ nm}$) This corroborates that State 2 corresponds to the starting point of the four-helix bundle, i.e., +8 layer. Our observation is also consistent with the report by (Ma et al, 2015) which suggests that the transition from State 2 to 3 corresponds to the unfolding of VAMP CTD (+2 to +8, 49-82 aa).

In conclusion, the Linker domain is separated to reach State 2, while the transition from State 2 to State 3 implies the unfolding of the CTD of VAMP from the four-helix bundle. Although there are many existing models/hypotheses to explain the interaction of Cplx and SNARE complex, our single molecule experiment is consistent with the major model (See our responses to the Point 9 for more details).

Figure R8. Complete configuration of the SNARE complex with and without Cplx. Fig. R8 (a) shows each domain in the SNARE complex. R8(b) shows the configurations of SNARE complex in state 1-5.

6. Figure 3 shows the unfolding force for Cpx mutant versus no Cpx. For the comparison, it also needs to show the force for full length Cpx. This data is not reported in earlier figures. The text refers for this to the Supplementary figures 5 A, B. However, the supplementary material has the figure 5 with a single panel which shows examples of force-extension curves but does not show the distribution of the unfolding forces.

Response: Thank you for your suggestion. We can see that adding full-length or 1-83 aa Cpx, the hopping takes place at ~21 pN in the absence of CpxI, while the average force increases to ~24 pN and some events happen at even higher force of ~43 pN. It is worth noting that there are more high-force signals in the full-length Cpx, which may imply more stable binding of full-length Cpx to SNARE complexes. We also added this figure as the supplementary figure S7.

Fig. R9. Unfolding force statistics of SNARE complex with full-length and 1-83aa Cpx.

7. What is "signal rate" (figs 4d, 5 c, and 6 a), and how is it related to anything? I was not able to find in the manuscript any information on how the "signal rate" was derived.

Response: Thank you for your precious comments and advice.

The fragments of or complete CpxI interacts with the SNARE complex, and the Extension trace of the SNARE complex may display three different states, i.e., the C-terminal stabilized state (CT),

middle clamped state (MC), and the C-terminal blocked state (CB). Here the probability to each state is defined as the signal rate,

$$\gamma_i = \frac{N_i}{N_{CT} + N_{MC} + N_{CB}}$$

$i = CT(C - \text{terminal stabilized state}), MC(\text{middle clamped state}), CB(C - \text{terminal blocked state})$

This definition has been used to process the signal rate in Fig. 4d, and Fig. 5c in the main text. Especially, there was only C-terminal inhibited state for CpxI 48-73aa in Fig 5a.

For we injected only buffer (first column in fig.6a) though protein channel as control, and the majority (81% of 31 molecules under test) showed no change after the injection. And once the fragment 83-134 aa (mostly part of CTD, without CH) was injected, and 18 of 27 SNARE complexes showed no change (second column of Fig. 6a), which was a negative control of experiments in Fig. 5. In Fig 6a, we define the signal rate as,

$$\gamma_i = \frac{N_i}{N_{Ct} + N_{MC} + N_{CB} + N_{NC}}$$

$i = CT(C - \text{terminal stabilized state}), MC(\text{middle clamped state}), CB(C - \text{terminal blocked state}), NC (\text{No change molecular})$

Those definitions were included in the update to the supplementary information in Section 6, and clarified this at Page 11 in the manuscript.

8. The distributions for the state occupancies (figs 2 c and 5 b) should be reported for every mutant. As is, these distributions are only shown for full length Cpx and for CTD mutant. What happened with all the mutants shown in figures 3 and 4?

Response: Thank you for your precious comments and advice.

In the revision, we had the chance to include them all in the supplementary Fig. Table 2. And also showed the open distances of four states at page 13 of the manuscript. Please also see the open distance for each of the mutants in the table below. The open distance was referenced to state 2, e.g., the open distance for state 2 is assumed to be 0. All the open distances are expressed in nanometer, while the numbers in the brackets expressed the standard deviation. The Average force records the average applied force on the SNARE complex when the trap separation keeps fixed.

	Average force[pN]	2	3	4	5
Without CpxI	20.16(0.78)	0	4.67(2.0)	7.02(2.3)	14.71(2.0)
FL C-terminal	19.50(0.62)	--	--	6.94(2.4)	14.25(2.0)
FL Middle-clamped	19.76(0.31)	--	4.9(1.6)	7.2(1.7)	--
FL Linker-open	20.47(0.72)	0	4.3(2.1)	--	14.1(2.4)
1-83aa	20.67(0.99)	0	5.97(2.2)	--	--
26-83 Linker-open	20.46(0.65)	0	6.2(2.2)	--	15.0(2.2)
26-83 C-terminal	21.15(0.97)	--	--	9.6(2.2)	15.0(1.9)
1-83+83-134 Linker-open	20.25(0.74)	0	4.9(2.3)	--	13.9(2.7)

1-83+83-134 (C-terminal)	18.65(0.67)	0	5.1(1.8)	8.1(1.9)	13.6(2.0)
1-83+83-134 (Middle-clamped)	18.53(0.57)	--	6.5(2.3)	10.0(2.2)	--

9. The model shown in Fig. 6 b should be discussed in relation to other models. The same relates to the results. For example, the 1-83 Cpx fragment drastically promotes SNARE assembly (Fig. 3a). This result seems to contradict a bunch of earlier studies, which showed that Cpx AH promotes the SNARE separation, not assembly. Of course, these direct single molecular studies are very powerful, and they may show that the earlier interpretations need to be reconsidered. But this discrepancy must be discussed.

Response: We thank the reviewer for the suggestion. The binding between center helix and SNARE is the prerequisite of the clamp function, the controversy focused on the mechanism of the interaction between accessory helix and the half-zippered SNARE. According to Rothman and colleagues' comprehensive mechanistic model, binding of the complexin central helix (amino acids 48–70) to the SNARE complex is a prerequisite for protein function, and interaction of the complexin accessory helix (amino acids 26–47) with the partly zippered SNARE complex inhibits complete C-terminal assembly and membrane fusion.

We summarized a bunch of models proposed in earlier studies in Fig. R10. These models include compete model, zigzag model, electrostatic repulsion model and direct-binding model [Fig.R10]. In the first zig-zag model, the central helix of complexin binds to one SNARE complex, while the adjacent accessory helix binds to a neighboring, second SNARE complex (Krishnakumar et al., 2011; Kummel et al., 2011). Therefore, the complexin may organize SNARE complexes into a zig-zag array—when interposed between vesicle and plasma membranes—that hinders fusion [Fig.R10a].

Based on the concentration of negatively charged amino acids within the accessory α -helix, Trimbuch and colleagues (Trimbuch et al., 2014) proposed a model, wherein this protein region inhibits release through enhancing electrostatic repulsion between vesicle and plasma membranes [Fig.R10b].

The accessory helix is thought to compete with the C-terminal portion of VAMP for binding to its cognate SNARE partners, hence providing an on-off switch by alternative zippering (Giraud et al., 2008) [Fig.R10c].

Another molecular mechanism for the accessory α -helix mediated clamp action has recently been proposed by (Bykhovskaia et al., 2013) using molecular dynamics simulation, they concluded that the accessory α -helix interacts directly with the v-SNARE vamp and thus arrests the zippering of the last hydrophobic layers +7 and +8 [Fig. R10d].

Our single molecule experiment on the interaction of various fragment of Cplx and SNARE complex provides single molecule support for the VAMP-binding model rather than the VAMP competition model. This is also consistent with the co-crystallization experiment, which suggests Cpx binds to the C-terminal of VAMP rather than complete.

Figure R10. Mechanism of complexin's clamp function. a. Zigzag model. b. Electrostatic repulsion model. c. Compete model. d. Direct-binding model.(Kummel et al., 2011; Mohrmann et al., 2015)

Experiment with mixer of Cplx fragments (83-134 aa and 1-83 aa) suggests the SNARE C-terminal becomes more blocked when the Cplx CTD (83-134 aa) increases, meanwhile the proportion of the C-terminal stabilized state decreases (Fig. 5c). Furthermore, the clamping function of Cpx can be reconstituted by fragment 1-83 aa of CpxI and its CTD as separate fragments in vitro (Fig. 5c, last two columns). Thus, physical continuity through the length of CpxI is not required to establish clamp function. CTD of CpxI inhibits the full zipper of SNARE complex, and plays an important role in the clamp function, in addition to CH being responsible for the direct combination of Cpx and SNARE, CTD is responsible for inhibiting the C-terminal assembly of SNARE complexes. At this time, it is quite reasonable for AH to be combined with the C-terminal VAMP far away from T-SNARE. At this point, the membrane fusion is in the priming state. At the arrival of the stimulus signal facilitating the membrane fusion, this clamp function of CTD is removed, CH still maintains the basic binding, AH can be assembled with VAMP C terminal closer to the SNARE complex, while NTD plays its role of facilitate, and together they stabilize the assembled SNARE complex with four spiral bundles. So that the assembled complex will not be immediately degraded by NSF, until the release of vesicle content is complete.

As we have not introduced the membrane and thus the electrical charge, our single molecule experiment can barely be used to explain the electrostatic repulsion theory. We agree with the reviewer that this is also interesting, and we are still working on how to introduce electrical charge to our system. Our single molecule study can provide potential insights to understand the zig-zag model in single molecule level.

Our results also **can't rule out** the Zig-Zag model, for the single molecular experiment set up is not suit for multiple molecular.

Minor:

1. Please state clearly what are "linker-open", "C-terminal-blocked", and "C-terminal-inhibited", and middle-clamped" states" – are they states 2, 5, 4, and 3 ? The table with names, numbers,

and separation distances for the states would help.

Response: We appreciate the reviewer to point out the expression of these terms. To avoid confusion and make them very clear, we have rechecked the whole manuscript and used consistent names for those states. All the states in the absence of Cplx are named with different numbers. Since the Cplx may interact with the SNARE complex, we mainly used the function of Cplx to name each binding state, i.e., “stabilized”, “blocked”, “inhibited”.

More detailed explanation and a concise figure to show all the configurations are given below.

In the absence of Cplx, The SNARE complex dynamically transits among 5 possible states depending on the magnitude of applied force. They are: State 1, SNARE complex fully folded; State 2, the linker domain open state; State 3, the C-terminal domain (CTD) open state; State 4, Middle Domain (MD) open state; State 5, N-terminal domain (NTD) open state. Those configurations are explicitly illustrated in the following figure.

Figure R11. Complete configuration of the SNARE complex with and without Cplx. a. shows each domain in the SNARE complex. Fig. 2(c) is reproduced from Fig. 2 in the main text for reference. b shows the configurations of SNARE complex in the absence (top) and presence (bottom) of Cplx.

In the presence of Cplx molecule, due to the interplay of Cplx and the SNARE complex, the Cplx molecule may bind to different domains on the SNARE complex, and thus displays different configurations of the Cplx-SNARE complex. Briefly, we interpret our single molecule data into three groups.

(1) **C-terminal-stabilized state:** The CplxI molecule preferentially bound on the C-terminal of the VAMP2 component in the SNARE complex. Therefore, the at a force range of 15 ~ 19 pN, the original SNARE complex keeps at State 2 with merely the linker domain open. The SNARE complex cannot unfold the linker domain (State 3), middle domain (State 4), and N-terminal domain (State 5). This has also been confirmed by our single molecule experiment that only State 2 appears in the single molecule traces in the presence of CplxI.

(2) **Middle-clamped state:** When the CTD in SNARE opens, the Cplx molecule may clamp the SNARE NTD and CTD domains with two domains in Cplx. As a result, the SNARE complex keeps the linker domain (LD) open, and can only hop between the middle domain (MD) folded and unfolded states.

(3) **C-terminal blocked state:** While the Cplx molecule only clamp the CTD of SNARE complex, the SNARE complex may hop among linker domain (LD) open state, middle-domain (MD) open state, and C-terminal (CTD) open state.

In summary, based on the explanation, the so-called “C-terminal stabilized”, “Middle-clamped”, and “C-terminal blocked” states demonstrate the function of Cplx molecule on the SNARE complex, while State 1 through 5 denote four different configurations of SNARE complex. They are associated but not exactly equal. It should be added that the C-terminal inhibited state only existed in 48-73(Figure 4a,4e), which we considered to be an incorrect/non-functional binding state, so it was not listed here.

In the updated manuscript, we have made them clear that States 1 through 5 represent the configuration of SNARE complex, while the above mentioned three states represent the interaction of Cplx on the SNARE complex. The map of the complete configuration is also included in the Supplemental Material (Supplemental Fig. S5).

2. All the interpretations and explanations should be moved to Discussion from Results. For examples, the paragraphs like:

“The optical tweezers were invented by A. Ashkin in 1986, who won the Nobel Prize 74 in Physics in 2018 because of the invention of optical tweezers and the application in 75 biology. The application of optical tweezers spans from colloidal physics, cell biology, to single molecule biophysics.”

Or:

“Structural studies can provide an insightful explanation. The crystal structures of Cpx: SNARE complex show different interaction surfaces when Cpx of different lengths is used. For Cpx 32-72 aa 23 or 24-73 aa 23), the binding position of Cpx CH corresponds to the middle part of the SNARE complex (0 ~ +1 layer); for a shorter Cpx 49-76 aa (containing only CH) 18 , the Cpx CH binds to the proximal C-terminal of the SNARE complex, When the Cpx peptide is too small, the binding position of CH moves towards the C-terminal of the SNARE complex. This suggests that if the Cpx fragment is too short (only CH), weak or even erroneous interactions are likely to occur, because the functional binding of Cpx requires the coordination of other Cpx domains. Our observation supports the hypothesis that CpxI AH-CH is the "minimum Clamp unit".

These should not be in the Results section.

Response: We appreciate your great suggestions, and have moved the interpretations and explanations to Discussion Section, we also updated the Discussion to make them more concise and readable. All the changes in the main text are marked in red.

3. The language needs improvement

Response: In the update, we have updated all the suggestions made by both reviewers, and spent efforts to went through the text to revise and polish the manuscript. We wish the reviewer will be satisfied with our update.

4. Figs S1-S3 are not cited in the right order

Response: We apologize for the mistake. In the revision, we have also supplemented more figures and tables. We have reordered and double checked all the figures and tables in both the main manuscript and the supplementary material, and updated all the citations to those figures and tables in both documents, including Figs S1-S3.

References in Responses to Reviewer's comments

- Bykhovskaia, M., Jagota, A., Gonzalez, A., Vasin, A., and Littleton, J.T. (2013). Interaction of the Complexin Accessory Helix with the C-Terminus of the SNARE Complex: Molecular-Dynamics Model of the Fusion Clamp. *Biophys J* *105*, 679-690.
- Choi, U.B., Zhao, M.L., Zhang, Y.X., Lai, Y., and Brunger, A.T. (2016). Complexin induces a conformational change at the membrane-proximal C-terminal end of the SNARE complex. *Elife* *5*.
- Gao, Y., Zorman, S., Gundersen, G., Xi, Z., Ma, L., Sirinakis, G., Rothman, J.E., and Zhang, Y. (2012). Single reconstituted neuronal SNARE complexes zipper in three distinct stages. *Science* *337*, 1340-1343.
- Giraud, C.G., Garcia-Diaz, A., Eng, W.S., Yamamoto, A., Melia, T.J., and Rothman, J.E. (2008). Distinct domains of complexins bind SNARE complexes and clamp fusion in vitro. *J Biol Chem* *283*, 21211-21219.
- Krishnakumar, S.S., Radoff, D.T., Kummel, D., Giraud, C.G., Li, F., Khandan, L., Baguley, S.W., Coleman, J., Reinisch, K.M., Pincet, F., *et al.* (2011). A conformational switch in complexin is required for synaptotagmin to trigger synaptic fusion. *Nature structural & molecular biology* *18*, 934-U998.
- Kummel, D., Krishnakumar, S.S., Radoff, D.T., Li, F., Giraud, C.G., Pincet, F., Rothman, J.E., and Reinisch, K.M. (2011). Complexin cross-links prefusion SNAREs into a zigzag array. *Nature structural & molecular biology* *18*, 927-933.
- Ma, L., Rebane, A.A., Yang, G., Xi, Z., Kang, Y., Gao, Y., and Zhang, Y. (2015). Munc18-1-regulated stage-wise SNARE assembly underlying synaptic exocytosis. *Elife* *4*.
- Mohrmann, R., Dhara, M., and Bruns, D. (2015). Complexins: small but capable. *Cellular and molecular life sciences : CMLS* *72*, 4221-4235.
- Schwartz, M.L., and Merz, A.J. (2009). Capture and release of partially zipped trans-SNARE complexes on intact organelles. *J Cell Biol* *185*, 535-549.
- Trimbuch, T., J., X., D., F., DR., T., J., R., and C., R. (2014). Re-examining how complexin inhibits neurotransmitter release.
- Yin, L., Kim, J., and Shin, Y.K. (2016). Complexin splits the membrane-proximal region of a single SNAREpin. *The Biochemical journal* *473*, 2219-2224.

Reviewers' comments:

Reviewer #1 (Remarks to the Author):

I think that the authors have adequately addressed my concerns and I suggest publication of the paper in its current form.

Reviewer #2 (Remarks to the Author):

The authors largely addressed my technical concerns, but the concerns about the interpretation remain

Major

1. The Table S2 shows rather small differences in open distances between the states 3 and 4. For the state 3, they show 5-6 Å, depending on the conditions, and for the state 4 it is 7-10 Å. In fact, the variation within the state 4 is larger than the difference between the states 3 and 4. Such a subtle difference between the states 3 and 4 (about 3 Å) would correspond to only one or two helical turns being unraveled. Yet the model (Fig. 1 d, Fig. S5, and fig 6b) shows a much more radical difference between the states 3 and 4. In my view, this is not supported by the data.

2. My concerns regarding the model were not addressed in the paper. The authors attempted to address them only in the rebuttal. The figure R10 in the rebuttal summarizes the existing models, as I suggested, but it is not clear how the obtained data agrees or disagrees with any of the models. Importantly, none of that was included in the paper. The paper still shows the author's model (Fig. 6b), and I do not see how it is justified by the data. This model disagrees with crystallography data and with all the existing models. If the authors indeed do believe that their model in 6 b is correct, they need a much stronger justification. Otherwise, they need to discuss how their data agree with the existing models or contradict them – currently, they attempted to do this in the rebuttal (Fig. R10) but not in the paper.

Minor

1. To show the mutations, the authors added sequences to Supplementary material (#14), but I still do not see the mutations. It is stated "red marked amino acids were mutated 283 amino acids" (line 282) but nothing is marked red. The mutations are shown in red in the rebuttal but not in the paper.
2. Fig. S7 needs to be labeled. I assume that red is cpx+ and blue is cpx- , but this needs to be shown in the figure or in the legend
3. Supplementary Table S2 needs units (I assume Angstroms)

Reviewer #1 (Remarks to the Author):

I think that the authors have adequately addressed my concerns and I suggest publication of the paper in its current form.

Response: We appreciate the reviewer for his/her support.

Reviewer #2 (Remarks to the Author):

The authors largely addressed my technical concerns, but the concerns about the interpretation remain

Response: We appreciate the reviewer for his/her support on the technical part. In the revision, we spent more efforts to interpret our single molecule data with the existing model. Please see our update marked in red.

Major

1. The Table S2 shows rather small difference in open distances between the states 3 and 4. For the state 3, they show 5-6 Å, depending on the conditions, and for the state 4 it is 7-10 Å. In fact, the variation within the state 4 is larger than the difference between the states 3 and 4. Such a subtle difference between the states 3 and 4 (about 3 Å) would correspond to only one or two helical turns being unraveled. Yet the model (Fig. 1 d, Fig. S5, and fig 6b) shows a much more radical difference between the states 3 and 4. In my view, this is not supported by the data.

Response: Thanks very much for the constructive comments of the reviewer, we revised our calculation of the corresponding protein structure of each state. We're sorry for the confusion and the unit should be "nm" in table S2.

To summarize, the reviewer has two major concerns: (1) The signal (open distance between states 3 and 4) is submerged in the noise; (2) the model is not supported by the data.

For Concern (1). Experimentally, the noise on the data trace would increase once the data acquisition frequency increases. Typically, in our single molecule experiment, the data are all collected at a sampling frequency of 20 kHz, and mean-filtered online to 10 kHz. The apparent noise in each state, for example 2.0 nm for State 3 in Line 2 in Table SR1, highly depends on the acquisition frequency. As we further filter the trace with larger average time, this frequency-dependent noise level would decrease. No matter how large the moving average filter is, the clear difference in States 3 and 4 do exist in the SNARE complex. This is also confirmed by the calculation according the model below in detail.

Practically, the data was first mean-filtered with average time of 0.2 ms to estimate the rough locations of each state, then feed into a more accurate HMM model to fit the experimental trace. Although the noise level shown in Table SR1 is very close to the open distance between State 3 and 4, this confirmed that our instrument can resolve tiny separation differences close to the noise limit. Note that, in some instances, State 2 disappear after the inclusion of Cpx molecules. The open distances for States 3, 4, 5 would be reference to the State 2 before/ without the inclusion of Cplx. In the revision we have marked those cases in the Supplementary table with "*".

For Concern (2). We have demonstrated that our measurement is consistent with the prediction from crystallographic structure. We would like to explain the consistency in more detail here.

Assume the contour length of the unfolded polypeptide is l_i , $i = 1, \dots, 5$, in different SNARE assembly states. $\alpha = 0.15/0.365 = 0.41$ is the ratio of the contour length of an amino acid (aa) in the helical conformation to that in the coil conformation. The extension of the unfolded polypeptide is related to the contour length and force through the Marko-Siggia formula. The

contour length difference for the unfolded polypeptide at this force point is evaluated based on the model shown in Fig. RS1 as,

$$\Delta l_i = \Delta x_i / r, \quad (1)$$

where Δx_i is the extension difference between the two states under the same tension F_i . The Δl is average contour length change of the unfolded polypeptide ($\Delta l_i = l_i - l_{i-1}$). The r is the extension-to-contour length ratio.

In the case of the α -helix, every 3.6 amino acid residues, the helix rises a circle, the pitch is 0.54nm, and the span of each residue is 0.15nm. When the polypeptide chain is fully extended, the length of each amino acid residue is 0.365 nm.

Figure SR1. Schematic model of the extension changes of the SNARE complex in its different assembly states.

Protein extension for state 1:

$$x_1 = l_1 r + h_0 \quad (2)$$

Protein extension for state 2:

$$x_2 = l_2 r + h_0 \quad (3)$$

Protein extension for state 5:

$$x_5 = l_5 r + (l_6 - l_5) \alpha \quad (4)$$

Protein extension for state 6:

$$x_6 = l_6 r \quad (5)$$

We could get the number of unfolded amino acids (N_i , $i = 1, 6$) is 34 aa and 147 aa and $h_0 = 2nm$ (the diameter of fully zippered four-helix-bundle) from our experiment design and the crystal structure of SNARE complex (Bracher et al., 2002; Chen et al., 2002). Therefore, the contour lengths of the unfolded polypeptides are that $l_1 = 12.41 \text{ nm}$, and $l_6 = 53.69 \text{ nm}$.

We experimentally measured a change of protein extension length $x_6 - x_1$ of 19.71 nm. Considering the known parameter h_0 , from Eques (2) and (5), we could calculate that $r = 0.526$

and $x_1 = 8.53 \text{ nm}$.

From the open distance of state 2-6 to state 1 in our experiment, we can further calculate the protein extension for each of those states: $x_2 = 11.53 \text{ nm}$, $x_3 = 16.2 \text{ nm}$, $x_4 = 18.55 \text{ nm}$, $x_5 = 26.24 \text{ nm}$, $x_6 = 28.24 \text{ nm}$. The number of unfolded amino acids ($N_i, i = 2,5$) in each state is 50, 100 correspondingly, therefore the contour lengths of the unfolded polypeptides are $l_2 = 18.12 \text{ nm}$, $l_5 = 36.44 \text{ nm}$ respectively.

Specifically, the Syntaxin unfolded 47 aa from state 5 to state 6. As Syntaxin contributed 49 aa in the state 2 (A254 to L205C), 2 aa unfolded between state 2 and state 3.

Since the crystal structure analysis only tackles the ensemble averaged conformations in the absence of mechanical force, the protein extension changes in two intermediate States 3 and 4 can only be inferred through the following,

Protein extension for state 3:

$$x_3 = l_3 r + (l_3 - l_2 - 2 \times 0.365) \alpha \quad (6)$$

Protein extension for state 4:

$$x_4 = l_4 r + (l_4 - l_3) \alpha \quad (7)$$

We can calculate $l_3 = 25.66 \text{ nm}$, $l_4 = 31.06 \text{ nm}$, which means that the number of unfolded amino acids ($N_i, i = 3,4$) in each state is 71 and 85 respectively. We could get the amino acids sites of state 3, 4 in the Fig. SR1, in which VAMP unfolded 19 aa (L84 to D65) from state 2 to state 3, corresponding to the unzipping of +8 layer to +3 layer in VAMP (+2 layer remain folded, Fig. SR2, state 3). For state 3 to 4, VAMP unfolded 14 aa (D65 to D51), corresponding to the unzipping of +2 layer to -1 layer in VAMP (-2 layer remain folded, Fig. SR2, state 4).

Figure SR2. Schematics of unfolding conformations of SNARE complex in states 2-4.

In conclusion, the open distances in our experiment are consistent with the crystal structure of the SNARE complex. We also updated all relevant models in the manuscript and the supplementary material, especially, we have updated the analysis in Section 18 in Supplementary material, and the Fig. 1d, Fig. 2b, Fig. 3d, Fig. S5. Specifically, we appended the Fig. 6b (Fig. SR3) in the following for easier understanding.

2. My concerns regarding the model were not addressed in the paper. The authors attempted to address them only in the rebuttal. The figure R10 in the rebuttal summarizes the existing models, as I suggested, but it is not clear how the obtained data agrees or disagrees with any of the models. Importantly, none of that was included in the paper. The paper still shows the author's model (Fig. 6b), and I do not see how it is justified by the data. This model disagrees with crystallography data and with all the existing models. If the authors indeed do believe that their model in 6 b is correct, they need a much stronger justification. Otherwise, they need to discuss how their data agree with

the existing models or contradict them – currently, they attempted to do this in the rebuttal (Fig. R10) but not in the paper.

Response: We thank the reviewer for the pertinent and rigorous suggestions. In our previous response, we have summarized various models reported by many groups on the SNARE complex, including the compete model, zigzag model, electrostatic repulsion model and direct-binding model. Since the Zig-Zag model explains how multiple SNARE complexes form a complete network, and our single molecule experiment only studies individual SNARE complex, we have not included the Zig-Zag model in the revision. However, our single molecule study may further provide potential insights to understand the zig-zag model.

As the electrostatic repulsion model deals with the interaction of the membrane and the SNARE complex, our single molecule experiment can also not support electrostatic repulsion model.

Since single molecule experiment was performed in the buffer solution in the absence of membrane, in the revision, we only discussed two models related to single molecule experiment in Fig. 6 (b).

Figure SR3. Models of complexin's clamp function (updated in Fig. 6b). Left: CTD compete model. Right: AH direct-binding model. Bottom: AH compete model.

Briefly, our single molecule optical tweezers experiment suggests that the CpxI can introduce the “C-terminal stabilized state”, “middle-clamped state” and “C-terminal blocked state” (Figs. 1, 4 and 5).

In the C-terminal blocked state, the distribution of SNARE complexes changed from 2~5 states to 3~5 states (Fig. 2a-d, S5), which implies that the reversible assembly and disassembly is limited to N-terminal, and the transition of C-terminal is not allowed (C-terminal-blocked state).

In the middle-clamped state, the dynamic folding of SNARE complex only took place between 3~4 states at this situation (Fig. 2a, 2b, 2d, S6a). The SNARE complex can also maintain in C-terminal stabilized state after the addition of CpxI (C-terminal stabilized state, Fig. 2a, 2b, 2d). Accordingly, SNARE complexes have changed from hopping among 2-5 states to being maintained only in the state 2 with the addition of CpxI (Fig. S6b).

The 1-83 aa of CpxI can dominantly stabilize the four helix-bundle of SNARE complex at the C-terminal (Fig. 3, 1-83 aa in Fig. 6a). The probability for C-terminal blocked state increases once the NTD (1-26 aa) is removed (26-83 aa in Fig. 6a) accompanied by an inhibition of the C-terminal stabilized state. The CTD (83-134 aa) of CpxI, however, dominantly blocks the C-terminal of SNARE (83-134 aa, in Fig. 6a). 1-83 aa of CpxI is insufficient to block the C-terminal zippering of SNARE complex compared with full-length CpxI, for the lack of CTD.

Our single molecule experiment on the interaction of various fragment of CpxI and SNARE complex provides single molecule support for the AH binding model rather than the AH competition model. This is also consistent with the co-crystallization experiment, which suggests Cpx AH binds to the C-terminal of VAMP rather than compete.

The controversy between these two models (Fig. SR3, right and bottom) is on the binding site of CpxI AH. Experiment with full length suggest that the SNARE C-terminal can be stabilized/locked (clamp) after the addition of CpxI. And CpxI fragments 1-83 aa can only stabilize the C-terminal of SNARE complex. If the full length CpxI realized the clamp function (C-terminal blocked state) by AH compete model (Fig. SR3 bottom), the binding site by CpxI AH is on t-SNARE without VAMP C-terminal (Vc). When Vc is zippered back to the four-helix bundle, there is no binding site for CpxI AH in the C-terminal stabilized state.

In contrast, if the full length CpxI realized the clamp function (C-terminal blocked state) by AH direct binding model (Fig. SR3 right), the binding site by CpxI AH is on Vc. When Vc is zippered back to the four-helix bundle, there is still the binding site for CpxI AH in the C-terminal stabilized state. Moreover, for the clamp function, we suggested the CpxI CTD insert the C-terminal of SNARE complex to compete with Vc or t-SNARE (prefer t-SNARE, (CTD compete model in Fig. SR3 left)) instead of CpxI AH (AH compete model in Fig. SR3 bottom), which is supported by our data mentioned below.

Experiment with mixture of CpxI fragments (83-134 aa and 1-83 aa) suggests the SNARE C-terminal becomes more blocked when the CpxI CTD (83-134 aa) increases, meanwhile the proportion of the C-terminal stabilized state decreases (Fig. 5c). Furthermore, the clamping function of Cpx can be reconstituted by fragment 1-83 aa of CpxI and its CTD as separate fragments in vitro (Fig. 5c, last two columns). Thus, physical continuity through the length of CpxI is not required to establish clamp function. CTD of CpxI inhibits the full zipper of SNARE complex, and plays an important role in the clamp function, in addition to CH being responsible for the direct combination of Cpx and SNARE, CTD is responsible for inhibiting the C-terminal assembly of SNARE complexes. Meanwhile, it is quite reasonable for AH to be combined with the C-terminal VAMP which is far away from t-SNARE. At this point, the membrane fusion is in the priming state. At the arrival of the stimulus signal facilitating the membrane fusion, this clamp function of CTD is removed, CH still maintains the basic binding, AH can be assembled with VAMP C terminal closer to the SNARE complex, while NTD plays its role of facilitate, and together they stabilize the assembled SNARE complex with four spiral bundles. So that the assembled complex will not be immediately degraded by NSF, until the release of vesicle content is complete.

In the revision, we have updated the analysis in discussion, and the related figures 6b in the manuscript.

All in all, our single molecule analysis supports the AH direct binding model, and rejects the AH competing model, then proposal our CTD compete model, all of which have close relationship with our experimental condition for in vitro single molecule experiment.

Minor

1. To show the mutations, the authors added sequences to Supplementary material (#14), but I still do not see the mutations. It is stated "red marked amino acids were mutated 283 amino acids" (line 282) but nothing is marked red. The mutations are shown in red in the rebuttal but not in the paper. Response: We are sorry for this mistake, and in the update, we have kept the mutated 283 amino

acids in red, and checked all other labels with color to make them appropriate and consistent with the text.

2. Fig. S7 needs to be labeled. I assume that red is cpx+ and blue is cpx-, but this need to be shown in the figure or in the legend

Response: We appreciate the reviewer for pointing out the missing of the label. In the update, we have included the label in the both the figure and the figure caption.

3. Supplementary Table S2 need units (I assume Angstroms)

Response: We have updated the units in Supplementary Table S2. The correct unit is nm (nanometer).

References

Bracher, A., Kadlec, J., Betz, H., and Weissenhorn, W. (2002). X-ray structure of a neuronal complexin-SNARE complex from squid. *J Biol Chem* *277*, 26517-26523.

Chen, X.C., Tomchick, D.R., Kovrigin, E., Arac, D., Machius, M., Sudhof, T.C., and Rizo, J. (2002). Three-dimensional structure of the complexin/SNARE complex. *Neuron* *33*, 397-409.

REVIEWERS' COMMENTS:

Reviewer #2 (Remarks to the Author):

The authors addressed my concerns, and I believe the paper is acceptable for publication